# QUASI MONTE CARLO METHODS ENABLE EXTREMELY LOW-DIMENSIONAL DEEP GENERATIVE MODELS

**Miles Martinez**[1,2,*]**, Alex H. Williams**[3,4,⋄]

[1] Department of Electrical and Computer Engineering, Duke University
[2] Center for Cognitive Neuroscience, Duke University
[3] Center for Neural Science, New York University
[4] Center for Computational Neuroscience, Flatiron Institute
*miles.martinez@duke.edu, ⋄alex.h.williams@nyu.edu

## ABSTRACT

This paper introduces *quasi-Monte Carlo latent variable models* (QLVMs): a class of deep generative models that are specialized for finding extremely low-dimensional and interpretable embeddings of high-dimensional datasets. Unlike standard approaches, which rely on a learned encoder and variational lower bounds, QLVMs directly approximate the marginal likelihood by randomized quasi-Monte Carlo integration. While this brute force approach has drawbacks in higher-dimensional spaces, we find that it excels in fitting one, two, and three dimensional deep latent variable models. Empirical results on a range of datasets show that QLVMs consistently outperform conventional variational autoencoders (VAEs) and importance weighted autoencoders (IWAEs) with matched latent dimensionality. The resulting embeddings enable transparent visualization and *post hoc* analyses such as nonparametric density estimation, clustering, and geodesic path computation, which are nontrivial to validate in higher-dimensional spaces. While our approach is compute-intensive and struggles to generate fine-scale details in complex datasets, it offers a compelling solution for applications prioritizing interpretability and latent space analysis.

## 1 INTRODUCTION

Deep generative models are a mainstay of contemporary machine learning. In addition to synthesizing complex images, sounds, and sentences, many researchers have sought to use these models to identify human-interpretable representations of complex data distributions. For example, variational autoencoders (VAEs) have been applied in scientific analysis of motor kinematics (Luxem et al., 2022), vocal communication patterns (Goffinet et al., 2021), single cell gene expression (Ding et al., 2018; Grønbech et al., 2020; Yan et al., 2022), neural population dynamics (Pandarinath et al., 2018), and extracellular spike waveforms (Beau et al., 2025).

Reducing dimensionality is seen as a key step towards interpretability in these models. VAEs achieve this by introducing a bottleneck layer. The size of this layer is a hyperparameter and often it is chosen to be modestly large. For VAEs, typical values are in the range of 32-128 latent dimensions and reconstruction quality is reported to drop off for networks with fewer than ~10 dimensions (Mai Ngoc & Hwang, 2020). As a result, it is virtually universal practice to employ further dimensionality reduction methods like UMAP or t-SNE to visualize the latent space of a VAE. Many papers also apply analyses such as clustering to identify human-interpretable structure in the latent space (see e.g., Goffinet et al., 2021; Luxem et al., 2022; Peterson et al., 2024; Koukuntla et al., 2025). Even in moderately high dimensions, these post hoc analyses are non-trivial to tune and validate.

Here, we demonstrate that in many cases two or three dimensional latent spaces are sufficient for performant deep generative models. Contrary to conventional wisdom (see, e.g., Doersch, 2016), we find that numerical integration can be a viable strategy to marginalize over the latent variables. We use randomized lattice integration rules (Dick et al., 2022) to approximate the marginal likelihood. We call the resulting model a *quasi-Monte Carlo latent variable model*, or QLVM. Although our

method requires multiple passes through the decoder on each training iteration, it completely eliminates variational approximations that can be challenging to tune and optimize (see Rainforth et al., 2018; He et al., 2019; Dai et al., 2020; Kinoshita et al., 2023, and Fig. 3 in this paper). Ultimately, we find that this approach decisively outperforms VAEs with comparable architectures and can even rival the performance of VAEs with higher capacity latent spaces.

Despite the limitations of our approach (see §4.2), there are many advantages to learning extremely low-dimensional embeddings of high-dimensional datasets. We can, for example, use kernel density estimators—which suffer a curse of dimensionality in higher dimensions—to precisely characterize the aggregate posterior density over the latent space. We can also exhaustively characterize the smoothness of the embedding by querying the Jacobian of the decoder network over a dense grid of points. We demonstrate how this information can be used to aid clustering analyses in §3.3, but the potential applications extend much more broadly to other topological analyses (Wasserman, 2018). Altogether, our results highlight a simple, but widely overlooked, approach to building extremely low-dimensional deep latent variable models for exploratory analysis.

## 2 METHODOLOGY

### 2.1 BACKGROUND ON DEEP LATENT VARIABLE MODELS

Let $\boldsymbol{x}_i \in \mathbb{R}^D$ for $i \in \{1, \ldots, n\}$ denote a sequence of high-dimensional, observed data points. Deep latent variable models seek to explain such datasets through the following probabilistic model:

$$\boldsymbol{x}_i \sim p_\theta(\boldsymbol{x}_i \mid \boldsymbol{z}_i) \qquad \text{where} \qquad \boldsymbol{z}_i \sim p(\boldsymbol{z}) \tag{1}$$

independently for $i \in \{1, \ldots, n\}$. Here, a $d$-dimensional latent variable vector $\boldsymbol{z}_i$ is drawn from a prior distribution $p(\boldsymbol{z})$.[1] The likelihood of each observation is parameterized by a deep network, called the *decoder*, with trainable weights $\theta$. We use $f_\theta(\boldsymbol{z}_i) = \mathbb{E}[\boldsymbol{x}_i \mid \boldsymbol{z}_i]$ to denote the mean output of the decoder. Ideally, the decoder weights would be trained to maximize the marginal likelihood:

$$p_\theta(\boldsymbol{x}_1, \ldots, \boldsymbol{x}_n) = \prod_{i=1}^n p_\theta(\boldsymbol{x}_i) = \prod_{i=1}^n \int p_\theta(\boldsymbol{x}_i \mid \boldsymbol{z}_i) p(\boldsymbol{z}_i) \mathrm{d}\boldsymbol{z}_i \tag{2}$$

Computing the marginal likelihood of a datapoint requires integrating over all possible values of $\boldsymbol{z}_i$. This integration is challenging in high dimensions and the simple-minded Monte Carlo estimate,

$$p_\theta(\boldsymbol{x}_i) = \int p_\theta(\boldsymbol{x}_i \mid \boldsymbol{z}_i) p(\boldsymbol{z}_i) \mathrm{d}\boldsymbol{z}_i \approx \frac{1}{m} \sum_{j=1}^m p_\theta(\boldsymbol{x}_i \mid \boldsymbol{z}_i = \widetilde{\boldsymbol{z}}_j) \tag{3}$$

where $\widetilde{\boldsymbol{z}}_1, \ldots, \widetilde{\boldsymbol{z}}_m$ are independent samples from $p(\boldsymbol{z})$, is therefore typically dismissed (Doersch, 2016; Mattei & Frellsen, 2018; Kingma et al., 2019). However, the quality of this approximation only deteriorates in proportion to the variance of $p_\theta(\boldsymbol{x} \mid \boldsymbol{z})$ for $\boldsymbol{z} \sim p(\boldsymbol{z})$ (see Owen, 2013). Thus, for sufficiently simple datasets, it is plausible that eq. (3) may suffice as a criterion to train $\theta$. To our knowledge, this possibility has not been seriously explored in deep latent variable models.

Since computing $p_\theta(\boldsymbol{x}_i)$ is often difficult, Variational Autoencoders (VAEs; Kingma & Welling 2014) instead target a lower bound on $\log p_\theta(\boldsymbol{x}_i)$. To do this, VAEs introduce a second deep neural network with parameters $\phi$ called the *encoder*. For each datapoint the encoder outputs a conditional distribution $q_\phi(\boldsymbol{z} \mid \boldsymbol{x}_i)$, which acts as a *variational approximation* to the posterior distribution of $\boldsymbol{z}_i$ given $\boldsymbol{x}_i$. As a training objective, one considers:

$$\mathcal{L}_{\mathrm{ELBO}}^{(i)}(\theta, \phi) = \frac{1}{m} \sum_{j=1}^m \log p_\theta(\boldsymbol{x}_i \mid \widetilde{\boldsymbol{z}}_j, \theta) - \mathrm{D}(q_\phi(\boldsymbol{z} \mid \boldsymbol{x}_i) \parallel p(\boldsymbol{z})) \tag{4}$$

where $\widetilde{\boldsymbol{z}}_1, \ldots, \widetilde{\boldsymbol{z}}_m$ are independent samples from $q_\phi(\boldsymbol{z} \mid \boldsymbol{x}_i)$ and $\mathrm{D}(\cdot \parallel \cdot)$ denotes the Kullback-Leibler (KL) divergence. In the limit that $m \to \infty$, this converges to a lower bound on $\log p_\theta(\boldsymbol{x}_i)$ called the *evidence lower bound* (ELBO). Typically, $m = 1$ sample is used to obtain an unbiased estimate of the lower bound. For further background see Doersch (2016); Kingma et al. (2019).

---

[1]Following common practice in this literature, we use the same notation for both the probability distribution of a random variable and its associated density function. We also refer to the two interchangeably.

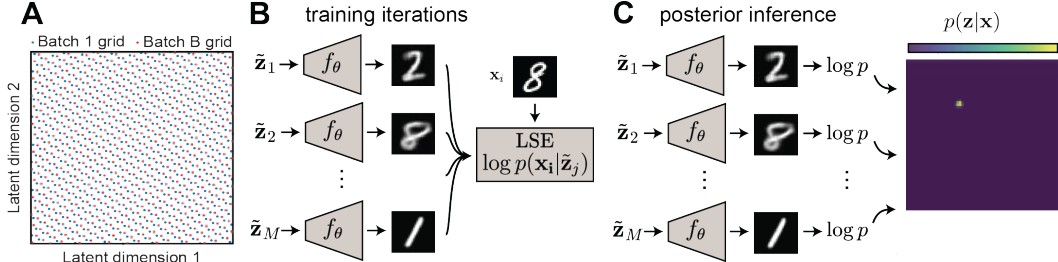

Figure 1: *(A)* On each batch, latent samples are generated by uniformly shifting a lattice of $M$ points (Batch 1 grid and batch B grid denote examples of samples that may appear during training). *(B)* These samples are fed through a shared decoder, $f_\theta$, to produce reconstructed datapoints (e.g. MNIST digits). During training, a log-sum-exp (LSE) reduction operator is applied to a vector of log probabilities, $\log p = \log p(\boldsymbol{x}_i \mid \tilde{\boldsymbol{z}}_j)$, to calculate eq. (5) up to a constant. *(C)* To approximate the latent posterior, $p(\boldsymbol{z} \mid \boldsymbol{x}_i)$, the same vector of log probabilities is normalized to form a discrete approximation over the $m$ lattice points. This employs Bayes' rule as described in the main text.

## 2.2 QUASI-MONTE CARLO LATENT VARIABLE MODELS (QLVMS)

VAEs target a lower bound on the marginal log likelihood, which is only tight when $q_\phi(\boldsymbol{z} \mid \boldsymbol{x}_i)$ matches the true posterior distribution of $\boldsymbol{z}$ given $\boldsymbol{x}_i$. Commonly, $q_\phi$ is assumed to follow a simple form—e.g. Gaussian with diagonal covariance. In very low-dimensional latent spaces, we will see that this assumption can be violated and, even if it approximately holds, the true posterior can be so narrow that it is difficult for the encoder network to learn (see Fig. 3).

We therefore sought to understand whether the simple Monte Carlo approach shown in eq. (3) can be practical in situations of interest. By taking logarithms on both sides of eq. (3), we obtain:

$$\mathcal{L}_{\text{MC}}^{(i)}(\theta) = \log p_\theta(\boldsymbol{x}_i) = \log \left[ \frac{1}{m} \sum_{j=1}^{m} p_\theta(\boldsymbol{x}_i \mid \tilde{\boldsymbol{z}}_j) \right] \tag{5}$$

where $\boldsymbol{x}_i$ is any datapoint and $\tilde{\boldsymbol{z}}_1, \ldots, \tilde{\boldsymbol{z}}_m$ are sampled points in the latent space. This is a lower bound on the marginal log likelihood in expectation. Indeed, by Jensen's inequality:

$$\mathbb{E}\left[ \log\left[ \tfrac{1}{m} \sum_j p_\theta(\boldsymbol{x}_i \mid \tilde{\boldsymbol{z}}_j) \right] \right] \leq \log\left[ \mathbb{E}\left[ \tfrac{1}{m} \sum_j p_\theta(\boldsymbol{x}_i \mid \tilde{\boldsymbol{z}}_j) \right] \right] = \log p_\theta(\boldsymbol{x}_i) \tag{6}$$

where the expectation is taken over $\tilde{\boldsymbol{z}}_1, \ldots, \tilde{\boldsymbol{z}}_m$. The inequality becomes tight as $m \to \infty$, since $\mathbb{V}\text{ar}[\tfrac{1}{m} \sum_j p_\theta(\boldsymbol{x}_i \mid \tilde{\boldsymbol{z}}_j)] \to 0$ in this limit (see Struski et al., 2023, for a detailed analysis). Thus, we achieve a more accurate estimate of the log marginal likelihood as we increase the computational budget (i.e. sample size) of our method. Moreover, the final equality in eq. (6) holds so long as the marginal distribution of each $\tilde{\boldsymbol{z}}_j$ is $p(\boldsymbol{z})$. This observation motivates quasi-Monte Carlo (QMC) methods, which jointly sample the points $\tilde{\boldsymbol{z}}_1, \ldots, \tilde{\boldsymbol{z}}_m$ to encourage uniform spacing while preserving the appropriate marginal distribution for each sample (L'Ecuyer, 2016; Owen, 2023).

To our knowledge, QMC has only been considered by a handful of researchers in the context of deep generative models. Buchholz et al. (2018) applied this approach to reduce the variance of ELBO estimates—i.e., to reduce variance in eq. (4) instead of eq. (5). More recently, Andral (2024) applied randomized QMC to generative models based on normalizing flows—a class of models which is poorly suited to our motivating application of nonlinear dimensionality reduction.

In our experiments, we set $p(\boldsymbol{z}) = 1$, i.e. uniform over $[0, 1)^d$, and we sample $\tilde{\boldsymbol{z}}_1, \ldots, \tilde{\boldsymbol{z}}_m$ jointly as a randomly shifted lattice (Fig. 1A). For $d = 2$, we use *Fibonacci lattices*, which optimally tile the unit square under periodic boundary conditions (Breneis & Hinrichs, 2020). For $d = 3$ we use *Korobov lattices* (see Owen, 2023, ch. 16.1). Although lattice integration rules can be applied to non-periodic functions (see Dick et al., 2022, ch. 7), we focus on latent spaces with periodic boundaries. To impose this desired boundary condition, we fix $\boldsymbol{z} \mapsto (\sin \boldsymbol{z}, \cos \boldsymbol{z})$ as the first layer of the decoder. We call the resulting model a *quasi-Monte Carlo latent variable model*, or QLVM.

During training, we simply compute eq. (5) over the lattice, leveraging a log-sum-exp operation for numerical stability (Fig. 1B). Note that there is no need to train an encoder network. At evaluation

time, we generate samples from a QLVM in the usual fashion by sampling $\boldsymbol{z} \sim p(\boldsymbol{z})$ and feeding $\boldsymbol{z}$ through the decoder to evaluate $f_\theta(\boldsymbol{z})$. To project high-dimensional data into the low-dimensional embedding space, we appeal to Bayes' rule, which tells us that $p(\boldsymbol{z}_i \mid \boldsymbol{x}_i) \propto p_\theta(\boldsymbol{x}_i \mid \boldsymbol{z}_i)$, due to the uniform prior on $\boldsymbol{z}_i$. We can therefore approximate the posterior as a normalized mixture of $m$ Dirac masses, with weights proportional to $p_\theta(\boldsymbol{x}_i \mid \widetilde{\boldsymbol{z}}_j)$ over lattice points (Fig. 1C). One can use the mean or mode of this approximate posterior as a low-dimensional embedding of $\boldsymbol{x}_i$.

**Changing the prior distribution.** As mentioned above, our experiments in §3 employ a latent space with periodic boundary conditions and a uniform prior. The uniform prior encourages points to be evenly spread and the periodic boundary conditions prevent embedded points from getting "stuck" in corners of the space. Additionally, lattice rules often have optimality properties for periodic functions (Breneis & Hinrichs, 2020), though they still work well for non-periodic functions.

It is very easy to adapt QLVMs to different sets of assumptions on the latent space. Indeed, a model uniform prior over the latent space is, in some sense, universal. Suppose that we wish to build a QLVM where $\boldsymbol{u}$ denotes the latent random variable and $p(\boldsymbol{u})$ denotes its prior. Under mild conditions, we can construct this as a one-to-one mapping $\boldsymbol{u} = \Phi^{-1}(\boldsymbol{z})$ where $\boldsymbol{z}$ is a uniform random variable and $\Phi^{-1}$ is the inverse CDF of $p(\boldsymbol{u})$. Thus, to construct a QLVM with a prior distribution $p(\boldsymbol{u})$, we can simply train a QLVM with a uniform prior over $\boldsymbol{z}$, but use $\boldsymbol{u} = \Phi^{-1}(\boldsymbol{z})$ as the input to the decoder. This approach is common in the QMC literature and is often called the *inversion method* (see Chpt. 4 in Owen 2013 and Chpt. 8 in Leobacher & Pillichshammer 2014). In appendix E.2 we show an example QLVM trained with respect to an isotropic Gaussian prior distribution on MNIST. This example model performs comparably well to a QLVM with a uniform prior.

## 2.3 RELATIONSHIP TO IMPORTANCE WEIGHTED AUTOENCODERS (IWAEs)

Importance weighted autoencoders (IWAEs, Burda et al., 2016) are a generalization of VAEs that target a different lower bound on the marginal log likelihood:

$$\mathcal{L}_{\text{IWAE}}^{(i)}(\theta, \phi) = \log \left[ \frac{1}{m} \sum_{j=1}^{m} \frac{p_\theta(\boldsymbol{x}_i \mid \widetilde{\boldsymbol{z}}_j) p(\widetilde{\boldsymbol{z}}_j)}{q_\phi(\widetilde{\boldsymbol{z}}_j \mid \boldsymbol{x}_i)} \right] \tag{7}$$

where $q_\phi$ represents the density of the proposal distribution (output by the encoder network) and $\widetilde{\boldsymbol{z}}_1, \ldots, \widetilde{\boldsymbol{z}}_m$ are independent samples from this distribution. One can show that the ELBO is recovered when $m = 1$. Further, so long as $q_\phi$ places nonzero density over the latent space, $\mathcal{L}_{\text{IWAE}}^{(i)}$ becomes, in expectation, a progressively tighter bound and approaches the true marginal log likelihood as $m \to \infty$ (Burda et al., 2016). Thus, IWAEs and QLVMs enjoy the same asymptotic theoretical advantages over standard VAEs. They also suffer similar computational drawbacks—in both, the quality of the lower bound comes at the cost of multiple forward and backward passes through the decoder. In IWAEs, multiple backward passes through the encoder are also required.

IWAEs are closely related to QLVMs. The major difference is that IWAEs use an encoder to learn a proposal distribution, $q_\phi$, while QLVMs draw samples from a fixed distribution—the prior. In this sense, QLVMs are a special case of an IWAE with a fixed proposal. Below we summarize two key reasons why fixing the proposal distribution may be advantageous from a practical perspective.

**Learning Good Proposals is Hard.** A bad proposal can lead to a lower bound with extremely high variance (see Owen, 2013, Ch. 9). It is costly to monitor the variance of the IWAE lower bound during training and unclear how to intervene to fix problems that arise. A second, less obvious, challenge to training IWAEs was revealed by Rainforth et al. (2018), who showed that increasing $m$ can simultaneously improve optimization of the decoder weights, $\theta$, while *deteriorating* the optimization of encoder weights, $\phi$. Roughly speaking, this occurs because the IWAE bound becomes more insensitive to the proposal as $m \to \infty$, and the magnitude of the gradient signal is averaged away quicker than the variance is reduced. See Tucker et al. (2018); Nowozin (2018); Morningstar et al. (2021) for further discussion.

**Computational Benefits of QLVMs.** QLVMs avoid the need to learn an encoder network. This saves GPU memory (since encoder weights and activations don't need to be stored), computation time (since gradients don't need to be backpropagated through the encoder), and hyperparameter sweeps (since the encoder network and optimizer don't need to be tuned). In addition to all of

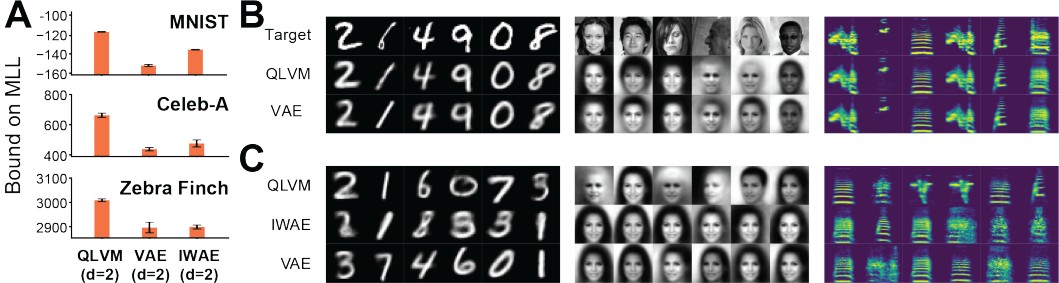

Figure 2: *(A)* Marginal log likelihood estimates on heldout test data from 2D models (orange) across three datasets (QMC estimate for QLVMs, ELBO for VAE and IWAE). The QLVM bound is higher than the ELBO and IWAE bound across all 2D models. Error bars indicate 1 standard deviation across 10 random seeds. *(B)* Reconstructions for a 2-D QLVM and VAE, with the QLVM displaying higher-quality reconstructions. *(C)* Samples from the prior of a 2-D QLVM, VAE, and IWAE. QLVMs show greater sample quality and sample diversity.

these benefits, training with minibatches is much more efficient with QLVMs because the randomly shifted lattice of latent points (see Fig. 1) can be re-used across all elements in the minibatch. There is no simple way to re-use samples in an IWAE because, for two datapoints $x$ and $x'$, the proposal distributions $q_\phi(z \mid x)$ and $q_\phi(z \mid x')$ may be very different. Thus, for a given batch size, $b$, and total budget of latent samples, $s$, an IWAE will be constrained to use $m = s/b$ samples per datapoint, whereas a QLVM can utilize $m$ latent samples to estimate the loss for each datapoint.

## 3 RESULTS

### 3.1 QLVMs PERFORM FAVORABLY IN VERY LOW-DIMENSIONAL SETTINGS

As discussed in §2, prior work regards the (quasi-)Monte Carlo estimator used by QLVMs as impractical. Here, we show that: *(a)* QLVMs outperform VAEs and other deep latent variable models when the latent space is constrained to be two-dimensional, *(b)* QLVMs' performance is better than one might expect in this setting across a variety of non-trivial benchmarks, and *(c)* in low-dimensional settings, the lower bounds targeted by standard VAEs and IWAEs are often loose.

We fit models to two image datasets (**MNIST** and grayscale **Celeb-A**), an acoustic library of bird song syllables (from Goffinet et al., 2021), and an acoustic library of Mongolian gerbil vocalizations (from Peterson et al., 2024). We compared two-dimensional QLVM models to two-dimensional VAE and IWAE models. Experimental details including preprocessing and network architectures are described in §B. Importantly, we used the same decoder architecture for all models to ensure fair comparisons.

In all cases, 2D QLVMs outperformed 2D VAE and 2D IWAE models both quantitatively in terms of reconstruction error (Fig. 2A), and qualitatively on example reconstructions (Fig. 2B). Moreover, novel samples found by sampling $z \sim p(z)$ and feeding $z$ through the decoder were, qualitatively, more diverse and of higher quality when derived from a QLVM than from VAE and IWAE models with matched latent dimensionality (Fig. 2C).

As one would expect, we found that VAEs with higher dimensional latent spaces often quantitatively outperformed 2D QLVMs—particularly on the most rich and challenging dataset of **Celeb-A** images. We show comparisons between 2D QLVMs and VAEs with various latent dimensionalities in appendix G. On simple datasets, such as **MNIST** or the zebra finch dataset, the improvements of higher-dimensional VAEs are qualitatively minor albeit still quantitatively detectable. In short, while 2D QLVMs struggle to capture fine-scale detail in complex datasets (e.g. high resolution natural images), they perform "good enough" on many other datasets of interest to scientists and engineers. Thus, QLVMs are an attractive method when 2D visualizations are desired.

Finally, we investigated the underlying reasons why VAE and IWAE models underperform in extremely low-dimensional latent spaces. We hypothesized that the encoder network struggles to iden-

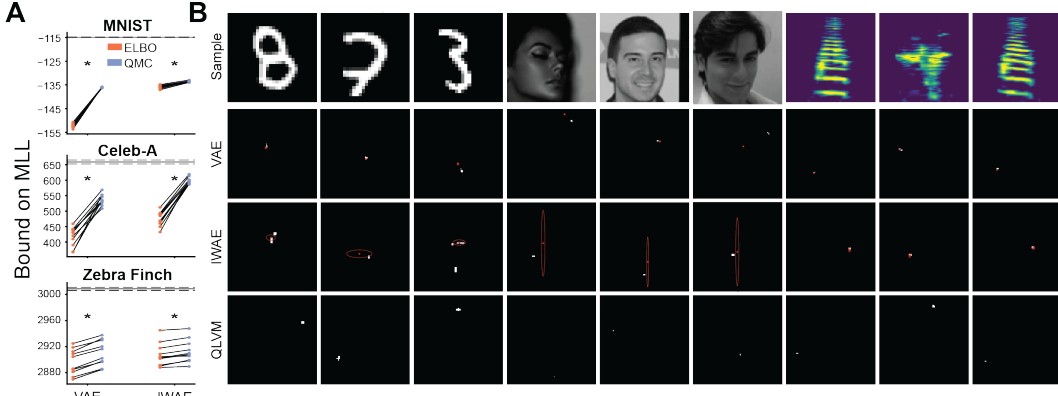

Figure 3: *(A)* VAE and IWAE lower bounds on marginal log likelihood (orange dots) for 2D latent spaces vs. the QMC estimate based on VAE and IWAE decoders (blue dots) from Fig. 2. Black horizontal line indicates the 2D QLVM bound on the marginal log likelihood. The gap between the blue dots and the black line represents the additional benefit of training with QMC samples. *(B)* Empirical vs. variational posteriors for **MNIST**, **Celeb-A**, and zebra finch vocalizations. Heatmap indicates 99% of probability mass in the empirical posterior (estimated by a dense lattice), while the red dots and ellipses indicate means and $\pm 3$ standard deviation of the variational posterior. This panel is best viewed on screen with magnification.

tify a good approximation to the latent posterior in these settings, resulting in a loose bound and a poor training signal for the decoder. To assess this, we took the decoders from trained 2D VAE and 2D IWAE models and applied the quasi-Monte Carlo integration rule in eq. (5) to get a tight estimate of the marginal log likelihood and the true posterior (using $m = 6724$ lattice points). This revealed that the bounds targeted by VAE and IWAE models were loose (Fig. 3A – blue dots are higher than orange, all $p < 0.05$ using one-sided binomial tests, Bonferroni corrected). Notably, tightening these bounds at evaluation time only did not immediately close the performance gap between VAE/IWAE baseline decoders and the QLVM decoder (Fig. 3A – blue dots below horizontal black line), demonstrating that our approach using the tighter QMC bound during training resulted in improved performance at evaluation time.

We then visualized these model posteriors as heatmaps, which qualitatively revealed that the VAE and IWAE variational approximations were often, though not always, poorly matched to the true posterior (Fig. 3B). Altogether, this analysis confirmed our suspicion that the encoder networks struggle to identify a good variational approximation over very low-dimensional latent spaces, motivating our approach to remove the encoder network entirely and instead use QMC to estimate the marginal log likelihood.

## 3.2 COMPUTATIONAL COST OF QLVMS

The quasi-Monte Carlo estimator used by QLVMs may appear to be computationally restrictive, even in low-dimensional spaces. For a fixed choice of decoder architecture, the tradeoff between QLVM performance and computational cost of training is mediated by the number of lattice points, $m$, over the latent space. The comparable parameter for IWAEs is $k$, the number of importance samples per data point in the batch. We therefore varied these hyperparameters and plotted model performance (negative ELBO for VAEs and IWAEs, negative QMC evidence bound for QLVMs) after 300 epochs, **MNIST** and zebra finch, or 10 epochs, **Celeb-A**) against computation time (the wall time to complete an epoch of training using one NVIDIA H100 GPU). This experiment traces out a 1D curve that represents the *Pareto front* of the QLVM model class (red lines in Fig. 4). Intuitively, any competing model that falls above or to the right of this curve is suboptimal to the QLVM—regardless of whether computational efficiency is prioritized over reconstruction error, or vice versa. Indeed, we observed that 2D VAE and 2D IWAE models with various architectures and hyperparameter settings consistently fell either above or to the right of the Pareto front, indicating underperformance.

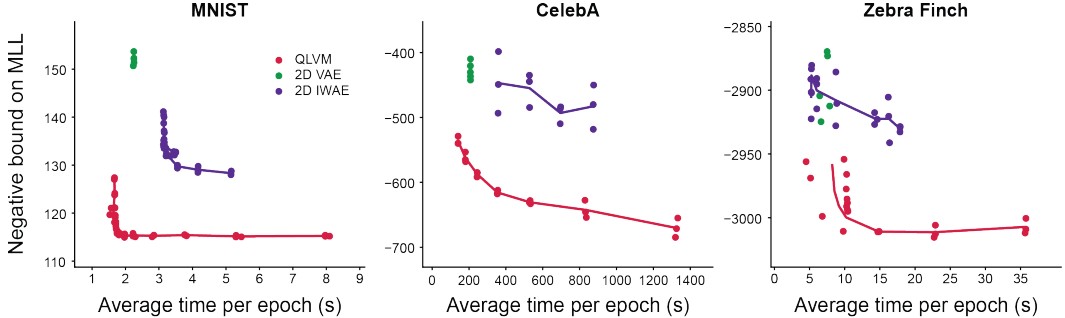

Figure 4: Performance vs. computational cost curves for 2D QLVMs (red), VAEs (green), and IWAEs (purple) on **MNIST**, **Celeb-A**, and zebra finch datasets. Curves closer to the lower left quadrant of each plot indicate a more favorable tradeoff. See §3.2 and appendix B.3 for comparison details.

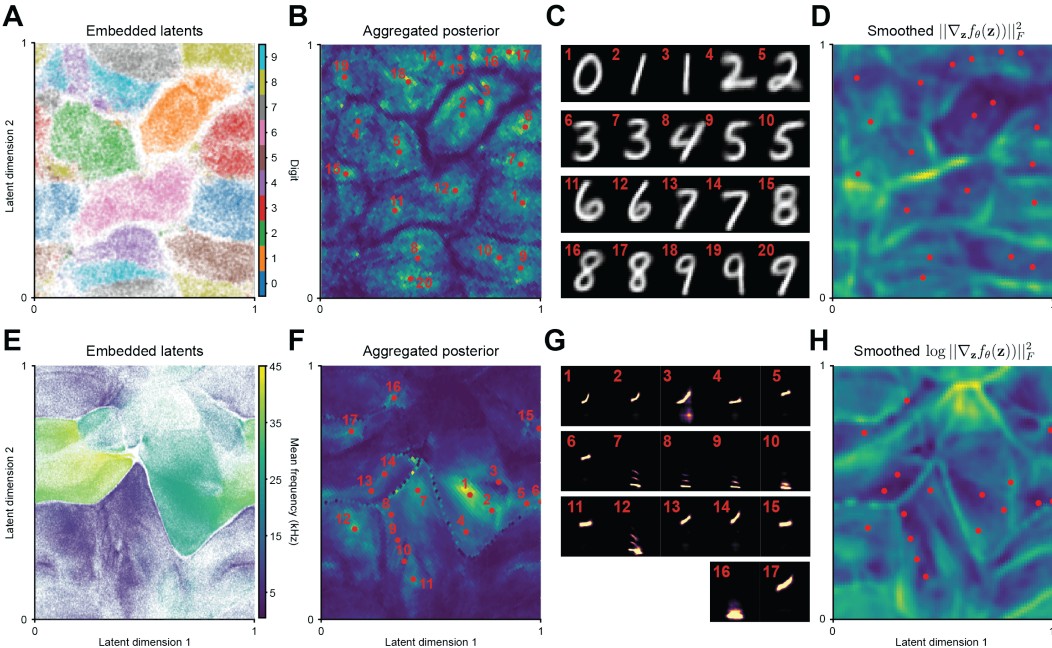

Figure 5: *(A)* Latent embeddings of **MNIST**, colored by digit. *(B)* Aggregated posterior of **MNIST** with mean-shift centroids (red) overlaid. *(C)* Reconstructions of centroids in *(B)*. *(D)* Smoothed Frobenius norm of decoder Jacobian on **MNIST**, with centroids from *(B)* overlaid. *(E)* Latent embeddings of gerbil vocalizations, colored by mean frequency of the vocalization. *(F)* Aggregated posterior of gerbil vocalizations, with mean-shift centroids (red) overlaid. *(G)* Reconstructions of centroids from *(F)*. *(H)* Smoothed log Frobenius norm of decoder Jacobian on gerbil vocalizations.

## 3.3 USING QLVMS FOR CLUSTERING

Clustering is a notoriously challenging, yet popular, form of unsupervised analysis. Identifying clusters is particularly difficult in high-dimensional spaces where conventional distance metrics (e.g. Euclidean) become less informative (Aggarwal et al., 2001). To address this, many works apply clustering algorithms on lower dimensional data representations that have been learned by a VAE or similar method (see e.g. Yan et al., 2022; Hadipour et al., 2022; Peterson et al., 2024). Since VAE performance is often compromised when the latent space is very low-dimensional, this approach still requires running a clustering algorithm in tens or hundreds of latent dimensions. Clustering is nontrivial in this setting, leading practitioners to introduce questionable parametric assumptions on the shape of each cluster (e.g. spherical in the case of K-means clustering).

To investigate the ability of QLVMs to help reveal clusters in high-dimensional datasets, we first examined 2D embeddings of the **MNIST** dataset. Using the procedure outlined in Fig. 1C, each datapoint was embedded into a 2D latent space and colored by the ground truth digit label (Fig. 5A). We also visualized the aggregated posterior density as a 2D histogram (Fig. 5B).[2] By visual examination, one can tell the clusters in Fig. 5A do not follow a simple or pre-specified shape. This motivates us to consider nonparametric clustering methods and related methods from topological data analysis (Wasserman, 2018). In Fig. 5C, we show the cluster centroids produced by the *mean-shift algorithm* (Cheng, 1995), whose locations are labeled in Fig. 5B. A single hyperparameter, *bandwidth*, controls the number of clusters (see appendix A.2). For illustrative purposes, we chose a relatively small bandwidth parameter to reveal finer-grained structure within each digit class.

We noticed that images with the same digit label naturally grouped together with small "gaps" separating each cluster (white space in Fig. 5A). We can show that these "gaps" correspond to regions where the map $z \mapsto f_\theta(z)$ defined by the decoder network changes rapidly. To reveal this, we queried the decoder Jacobian—i.e., the matrix containing gradients of each output dimension of $f_\theta(z)$ with respect to $z$. The Frobenius norm of this matrix is related to Fisher information (Berardino et al., 2017) and is a measure of the discriminability of nearby points in the latent space. We find that values of $z$ with large Jacobians present themselves at cluster boundaries (Fig. 5D), effectively highlighting the valleys of the 2D aggregate posterior in Fig. 5B. In Appendix C, we visualize latent space traversals confirming rapid changes in the decoded image. This visualization highlights a unique feature of QLVMs that is absent in methods like UMAP (which don't contain a generative process) as well as alternative generative models with higher-dimensional latent spaces (where "gaps" cannot be easily identified or visualized).

We then applied clustering in the QLVM space to an acoustic library of Mongolian gerbil vocalizations from Peterson et al. (2024). The number of vocal call types and the transition statistics between these calls is an open scientific question, making this dataset a more realistic and challenging testbed. We repeated the same set of analyses and visualizations as above. Embedded vocalizations segregated according to their mean frequency with visible "gaps" between putative clusters (Fig. 5E). The mean-shift algorithm identified a number of prototypical vocal call types (Fig. 5F-G), and (as in **MNIST**) the norm of the decoder's Jacobian was sharply peaked at the gaps between clusters (Fig. 5H), which corresponded to rapid changes in the decoded vocalization (appendix C). While further investigation is needed, these results suggest that the gerbil vocal repertoire can be coarsely categorized into low, middle, and high-frequency calls and that many of the finer-scale clusters (in Fig. 5G) roughly lie along a spectrum. Indeed, many of the fine-scale clusters are not separated by a clear boundary defined by the decoder's Jacobian (Fig. 5H) and it is possible to smoothly interpolate between many, though not all, of the fine-scale vocal clusters (Figs. A4 and A5).

## 3.4 USING QLVMS TO VISUALIZE CONTINUOUS LATENT DIMENSIONS

We explicitly constrain our latent space in these models to be low-dimensional. However, there are cases where the *true* latent dimensionality is higher than that of our model. One such case is captured by the **3dShapes** dataset (Burgess & Kim, 2018), in which high-contrast images are generated from 6 latent factors (wall hue, floor hue, object hue, object shape, object scale, and orientation). Importantly, *this dataset is constructed to have minimal clustering*—each latent factor, other than shape, is varied continuously. See Fig. A1 for example images from the dataset.

We trained a 2d QLVM on these data and compared the embedding to a 2d UMAP embedding (Fig. 6). UMAP artificially created small clusters of data, which are not representative of the true nature of the latent factors. QLVMs, on the other hand, represented each latent feature as a continuum across the latent space. Additionally, UMAP was able to separate one latent factor (floor hue) into unique clusters, but mixed all other latent factors beyond that. QLVMs smoothly encoded three of the six latent factors (wall hue, floor hue, object hue), which are responsible for the majority of variance in pixel space across images. The remaining three latent variables (shape type, scale, and orientation) were embedded in an increasingly non-smooth manner. Importantly and in contrast to the experiments in §3.3, the QLVM embedding showed little evidence of clustering—the aggregate posterior was uniform and the decoder Jacobian showed no sharp ridges across the latent space (see

---

[2]Note that the relatively uniform spread of the points across the latent space and periodic boundary conditions in Fig. 5 are expected due to the uniform prior and our construction of a periodic decoder (see §2.2).

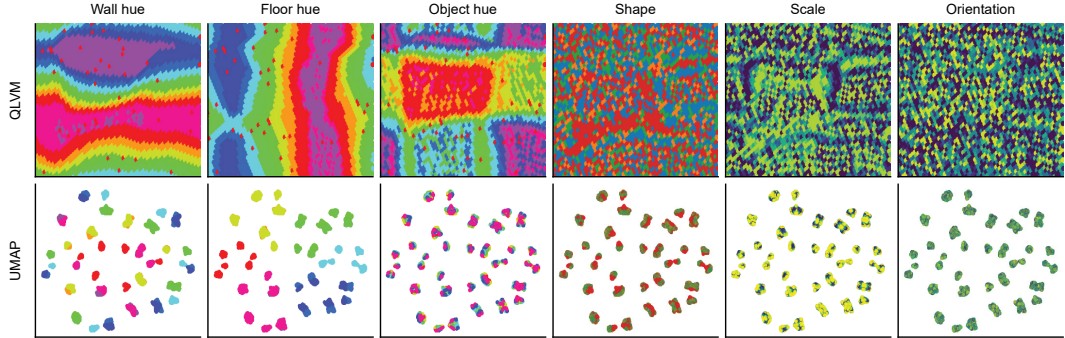

Figure 6: Embeddings of the 6d **3dShapes** dataset into 2 dimensions by a QLVM (top row) and UMAP (bottom row). For the QLVM, color indicates the maximum probability latent factor value at that point in the space. Wall, floor, and object hues are matched to the ground truth hue. Cube, cylinder, sphere, and pill are shown as blue, orange, green, and red under shape. Scale and orientation range from $[0, 1]$ and $[-30, 30]$ and corresponding colors range from dark blue to yellow.

appendix D). Together, these results demonstrate how QLVMs can nonlinearly pack additional information into a very low-dimensional space and highlight an advantage over UMAP, which is prone to hallucinate clusters (which has been previously noted by Zhai et al., 2022; Wang et al., 2021).

## 4 DISCUSSION

We introduced *quasi-Monte Carlo latent variable models*, or QLVMs—a simple but effective approach to achieve very low-dimensional embeddings based on a deep generative model. While these models struggle to capture finescale details in complex datasets (e.g. high-resolution images), they are an attractive option for practitioners who prioritize interpretability over sample quality. We view QLVMs as a strong alternative to popular methods like UMAP (McInnes et al., 2018), tSNE (Van Der Maaten, 2009), and ISOMAP (Tenenbaum et al., 2000), which are popular for learning 2d or 3d embeddings, but do not learn an explicit data generating process.

Because they can be directly visualized, QLVM embeddings are amenable to a variety of transparent manipulations and *post hoc* analyses such as clustering (§3.3), calculating geodesic paths (appendix C; Arvanitidis et al. 2018), and visualizing smoothness of the decoder mapping (Fig. 5D,H). One can easily imagine additional applications, such as estimating mutual information or penalizing the Lipschitz constant of the decoder network during training (see e.g. Kinoshita et al., 2023). All of these analyses quickly become nontrivial as additional latent dimensions are considered.

### 4.1 OTHER RELATED WORK

We focused on standard VAEs and IWAEs as competitors to QLVMs. Although these are the most well-established models, there is a broader literature on this subject including work that explores sequential and Markov Chain Monte Carlo methods (Salimans et al., 2015; Goyal et al., 2017; Huang et al., 2018; Thin et al., 2021), iterative refinements of the variational distribution output by the encoder network (Kim et al., 2018; Marino et al., 2018; Emami et al., 2021), and leveraging ensembles of models for improved importance sampling (Kviman et al., 2022). In general, these methods come with nontrivial hyperparameter choices such as the number of MCMC steps, stochastic transition kernels, and annealing schedules. Moreover, they often incur additional computational costs—for example, by fitting ensembles of encoder networks (in Kviman et al., 2022) or by introducing and optimizing backward Markov kernels (e.g. Salimans et al., 2015). It is noteworthy that Thin et al. (2021) motivate this general line of work by stating that IWAEs scale poorly to high-dimensional latent spaces, which is in direct contrast to our goal to identify very low-dimensional representations. In appendix F, we show a direct comparison between QLVMs and iterative amortized inference Marino et al. (2018) on 2D MNIST embeddings, confirming that the latter is considerably more time intensive to train and performs similarly to the IWAE baseline shown in the results section.

### 4.2 LIMITATIONS

**Sample quality.** Deep latent variable models are leveraged for a variety of applications in science and engineering. In certain settings, such as image or audio synthesis, the primary goal is to generate high-resolution, high-quality, and diverse samples. In these cases, state-of-the-art methods make use of high-dimensional and hierarchically structured latent variables (Vahdat & Kautz, 2020; Rombach et al., 2022). As noted in section 2, the quasi-Monte Carlo approximation employed by QLVMs does not scale well to high-dimensional spaces. Thus, our work has limited utility to practitioners who are willing to sacrifice interpretability for state-of-the-art performance and sample quality. However, QLVMs are appealing in applications where interpretability is prioritized.

**Scaling to high-dimensional latent spaces.** QLVM performance is driven by densely sampling the latent space with a discrete lattice of $m$ points. Increasing $m$ improves performance, but can be computationally costly. Fortunately, in low-dimensional spaces, these costs are not nearly as bad as one might imagine (see Fig. 4). In principle, IWAE models with expressive and well-trained encoders could approximate the marginal log likelihood with a smaller number of latent samples than a QLVM. However, this can be difficult to achieve in practice as seen both in previous work (Rainforth et al., 2018) and in our results in Fig. 4. Furthermore, as discussed in section 2.3, IWAEs suffer several computational drawbacks that are overcome by QLVMs. The main computational limitation of QLVMs is their ability to scale to high-dimensional latent spaces—due to the curse of dimensionality, the number of latent samples must increase exponentially with the latent dimension to maintain a similar density of sample points.

**Interpretability of low-dimensional latent spaces.** While very low-dimensional embeddings have advantages, they must also be interpreted with care. For example, the **3dshapes** dataset contains 6 "ground truth" features. These features must somehow be mixed together when we fit a 2D QLVM. Indeed, we find that these six factors follow smooth nonlinear profiles over the embedding space, as shown in Fig. 6. The embedding is clearly organized with respect to three of the six features (wall hue, floor hue, and object hue) but the patterns with respect to the remaining three features are not easy to visualize. Of course, it is impossible to linearly disentangle more than two features in a 2-dimensional latent space, so these limitations are not unique to QLVMs. Nonetheless, this is a broader limitation of visualization techniques that future research may wish to address.

**Identifiability.** The optimal embedding identified by deep latent variable models is generally not unique (Hyvärinen & Pajunen, 1999) unless further assumptions are introduced (Khemakhem et al., 2020). In their simplest form, QLVMs do not address this concern. Future work should investigate the extent to which this non-uniqueness impacts post-hoc analyses such as clustering.

### 4.3 POSSIBLE EXTENSIONS

One can imagine many refinements to the approach we've outlined in this paper. First, it would be interesting to explore whether some of the ideas discussed in §4.1 related to Monte Carlo might be incorporated to refine the QLVM objective. For example, one could use a lattice to initialize an adaptive importance sampling routine (Bugallo et al., 2017). Such improvements could enable the model to utilize the latent space at a finer scale than is specified by a fixed lattice, thereby improving the quality of generated samples and addressing a key limitation.

Another extension would be to incorporate additional domain knowledge as conditional variables that modulate the latent space—a principle that is already well-established in the broader literature (Sohn et al., 2015). In many applications, we expect this to aid model interpretability as well as expressivity. For example, similar body movements can be executed at faster or slower speeds; this motivated Wu et al. (2025) to use conditional VAEs to disentangle speed from unsupervised latent dimensions (see also Costacurta et al., 2022). Similarly, in acoustic datasets (e.g. in Fig. 5E-H), it would make sense to condition on features such as syllable duration, volume, and estimates of the fundamental frequency. We expect that providing the decoder with these statistics would also improve the quality of QLVM reconstructions, while preserving the interpretability of the embedding.

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

## A  ADDITIONAL MATHEMATICAL DETAILS

### A.1  LATTICES

Throughout, a randomly shifted lattice is defined modulo 1. That is, if $\boldsymbol{u}_1, \ldots \boldsymbol{u}_m$ define a $d$-dimensional lattice, we set $\widetilde{\boldsymbol{z}}_j = \boldsymbol{u}_j + \Delta - \lfloor \boldsymbol{u}_j + \Delta \rfloor$ where $\Delta \sim \text{Unif}([0,1]^d)$ is a random shift and $\lfloor \cdot \rfloor$ denotes the *flooring operation*.

A *lattice* is a set of points $L \subset \mathbb{R}^d, d \geq 1$ that

1. is closed under addition and subtraction ($x, y \in L \implies x + y \in L, x - y \in L$), and
2. contains no repeat points ($\forall x, y \in L, x \neq y, \exists \epsilon > 0$ such that $\|x - y\| > \epsilon$)

A *lattice rule* is a lattice $L$ constrained to the unit hypercube. In this work we use rank-1 lattice rules, which are lattice rules such that $\boldsymbol{u}_j = \frac{j\mathbf{b}}{m}$, $\mathbf{b} \in \mathbb{Z}^d$ — in particular, we use *Korobov* lattice rules, in which $\mathbf{b} = [1, a, a^2 - \lfloor \frac{a^2}{m} \rfloor, \ldots, a^{d-1} - \lfloor \frac{a^{d-1}}{m} \rfloor]$, $a \in \{2, 3, \ldots, m-1\}$. The Fibonacci lattice is a special case of Korobov lattices in which $d = 2$, $a = \text{Fib}(k-1), m = \text{Fib}(k)$, where $\text{Fib}(k)$ is the k-th element of the Fibonacci sequence. As Fibonacci lattice rules have been shown to optimally reduce integration error over the unit square, we use those for all two-dimensional latent spaces; we use more general Korobov lattice rules for three-dimensional latent spaces. See Dick et al. (2022); Owen (2023) for further details regarding lattice rules.

### A.2  MEAN-SHIFT CLUSTERING

The mean-shift algorithm is a mode clustering method defines clusters based on modes of density of the data. This algorithm begins with a set of seed points and finds modes by iteratively moving these seed points towards areas of high density in local neighborhoods, defined by a bandwidth parameter $h > 0$. Starting from $\boldsymbol{z}^{(0)}$, updates are defined as:

$$\boldsymbol{z}^{(j+1)} = \frac{\sum_i \boldsymbol{z}_i K\left( \frac{||\boldsymbol{z}_i - \boldsymbol{z}^{(j)}||}{h} \right)}{\sum_i K\left( \frac{||\boldsymbol{z}_i - \boldsymbol{z}^{(j)}||}{h} \right)} \tag{8}$$

where $K(\cdot)$ is a smoothing kernel; in our experiments we set $K(u) = \exp(-u^2)$. Given that our QLVM gives us a density estimate, we can replace points $\boldsymbol{z}_i$ with our lattice points, $\tilde{\boldsymbol{z}}_i$, and density estimates with our aggregate density $\mathbb{E}_{\boldsymbol{x}}[p(\boldsymbol{z}|\boldsymbol{x})]$. In a neighborhood $N_h$ around our current mode estimate:

$$\boldsymbol{z}^{(j+1)} = \frac{\sum_{\tilde{\boldsymbol{z}}_i \in N_h} \tilde{\boldsymbol{z}}_i \mathbb{E}_{\boldsymbol{x}}[p(\tilde{\boldsymbol{z}}_i|\boldsymbol{x})]}{\sum_{\tilde{\boldsymbol{z}}_i \in N_h} \mathbb{E}_{\boldsymbol{x}}[p(\tilde{\boldsymbol{z}}_i|\boldsymbol{x})]} \tag{9}$$

Upon convergence of all initial seeds to local modes, seeds within the set bandwidth of each other are merged, favoring seeds with the highest overall density. See (Wasserman, 2018) for further details.

## B  EXPERIMENTAL DETAILS

All models were trained to convergence using Adam with learning rate=0.001, $\beta_1 = 0.9$, $\beta_2 = 0.999$. **MNIST** and gerbil vocalizations were trained using Bernoulli log probability over pixels, all other datasets were trained using Gaussian log probability with variance= 0.1. For VAEs and IWAEs, variational posteriors were Gaussian distributions, latent priors were standard Gaussian distributions. QLVM training took between 30 minutes (**MNIST**) and 9 hours (**3dShapes**) on one NVIDIA H100 GPU. For all IWAEs we used $m = 10$ importance samples.

### B.1  DATASETS

#### B.1.1  **MNIST**, **Celeb-A**, AND **3dShapes**

For both **MNIST** (LeCun et al., 2010) and **Celeb-A** (Liu et al., 2015) we used default train/test splits. Celeb-A images were downsized to 80x80 pixels and converted to grayscale. For **3dShapes** (Burgess

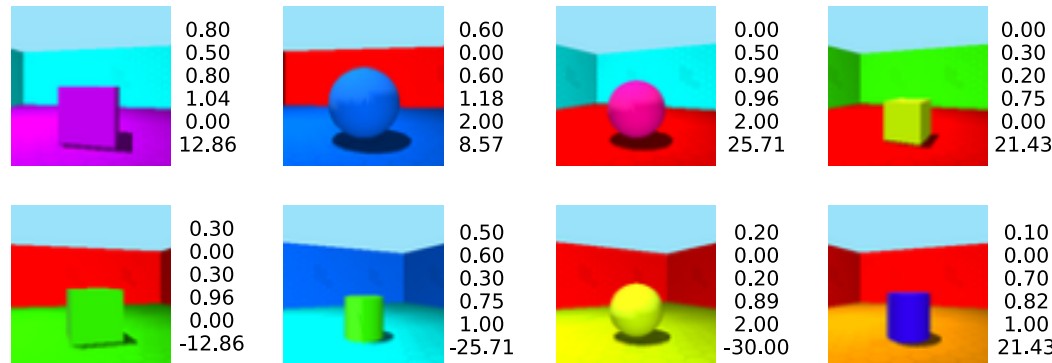

Figure A1: Example samples from the **3dShapes** dataset. Numbers indicate latent factor values. From top to bottom: Floor hue, wall hue, object hue, scale, object shape, orientation.

& Kim, 2018), models were trained using an 80/20 train/test split. See Fig. A1 for examples of the **3dShapes data**.

### B.1.2 ZEBRA FINCH AND GERBIL VOCALIZATIONS

We used vocalizations from a single adult zebra finch, Blu285 (Goffinet et al., 2021), and vocalizations from three different families of gerbils (Peterson et al., 2024). Vocalizations were segmented and processed into spectrograms using Autoencoded Vocal Analysis (Goffinet et al., 2021). Models on both datasets were trained using an 80/20 train/test split.

### B.1.3 CMU MOTION CAPTURE

We used motion capture data from the Carnegie Mellon University Motion Capture dataset (https://mocap.cs.cmu.edu/) — specifically, subject 54 (human pantomiming animal behaviors). Each trial consists of a sequence of poses; for each of the 44 non-torso joints in the pose we extract the sine and cosine of its Euler angle. We perform the same transformation to 3 global (torso) pose angles, and from these compute global (torso) translational velocities. Each data point fed to the model was a stack of 10 contiguous transformed frames; models were trained using an 80/20 train/test split.

### B.2 ARCHITECTURES

See Fig. A2 for all decoder architectures used for all experiments. VAE and IWAE encoders had the same structure as decoders in the reverse order, with *ReLU* nonlinearities replaced by hyperbolic tangent nonlinearities for stability during training.

### B.3 PERFORMANCE BY NUMBER OF SAMPLES IN QLVMS AND IWAES

To compare the computational cost (vs. model performance) in IWAEs and QLVMs, we trained each model on **MNIST**, **Celeb-A**, and the zebra finch dataset, varying the number of importance samples $k$ in IWAEs and number of lattice points $m$ for QLVMs. For IWAEs, we used:

- **MNIST**, $k \in [5, 10, 15, 20, 25, 30, 35, 40, 85, 140, 225]$
- **Celeb-A**, $k \in [5, 10, 15, 20]$
- zebra finch, $k \in [5, 10, 15, 30, 60, 70, 80]$

For QLVMs, we used:

- **MNIST**, $m \in [55, 89, 144, 233, 377, 610, 987, 1597, 2584, 4181, 6765, 10946, 17711, 28657, 46368]$
- **Celeb-A**, $m \in [55, 89, 144, 233, 377, 610, 987]$
- zebra finch, $m \in [55, 89, 144, 233, 377, 610, 987]$

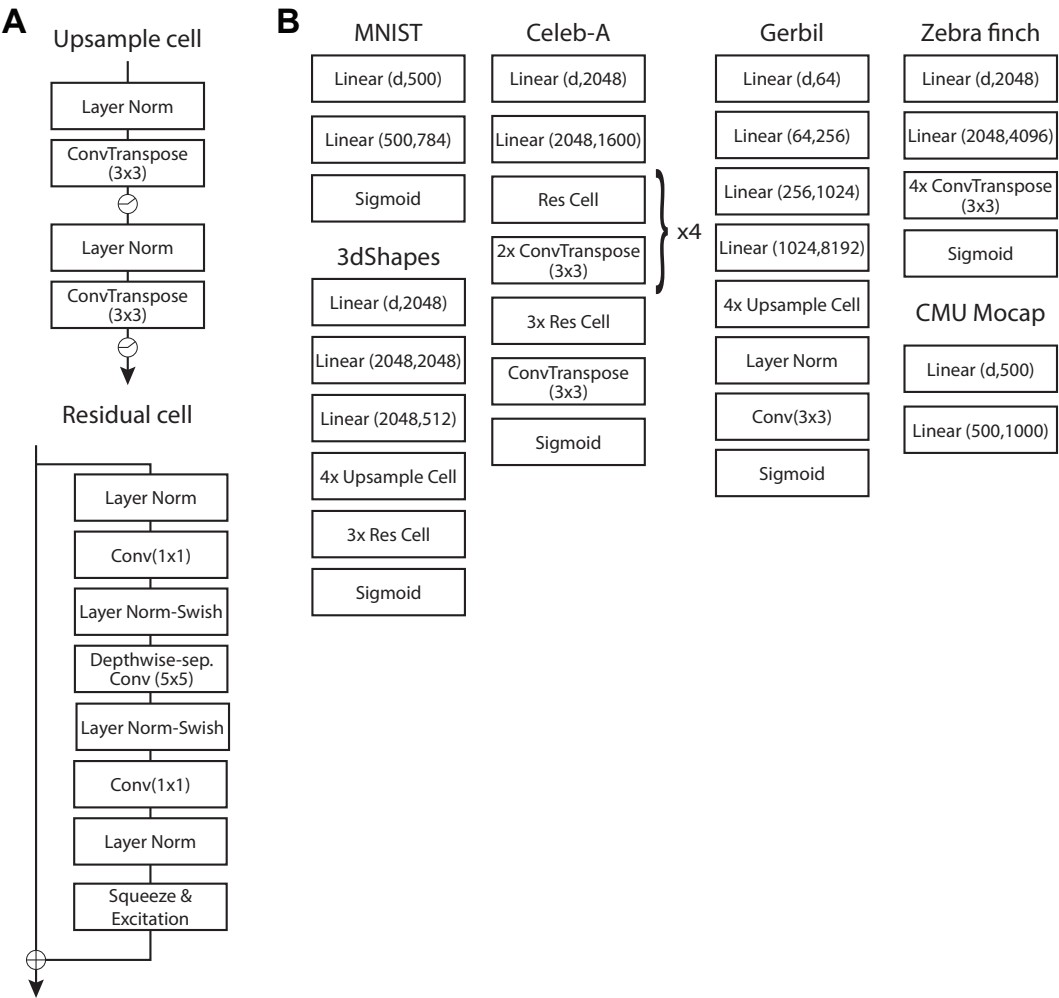

Figure A2: *(A)* Neural network cells used in models. *(B)* Model decoders used in all experiments. All linear and convolutional layers prior to the *sigmoid* output were followed by a *ReLU* nonlinearity.

Architectures were the same as those used in all other experiments; we trained three models for each value of $m$ and $k$.

## C LATENT TRAVERSALS AND GEODESICS

Because of the low dimensionality of our latent space, we can perform linear latent traversals (as typically carried out to explore dimensions of LVMs) or find geodesics based on the aggregate density over the data. Additionally, since our latent space is constrained to the unit $d$-cube, we can traverse over all values of each latent variable. We do so in Fig. A3 for each dataset. In each, we see that, for the most part, moving smoothly through latent space results in smooth changes to the output images. Abrupt changes occur at what appear to be class boundaries.

We can probe these regions of what seem to be abrupt change based either on reconstructions or on the aggregate posterior. We demonstrate one such case on the gerbil vocal dataset in Fig. A4. Traveling parallel to what appears to a region of low density results in little change in the output, while there is an abrupt change in the output spectrogram the moment that boundary is crossed.

Finally, we can compute geodesics connecting points in the latent space (Arvanitidis et al., 2018). Specifically, we construct a weighted graph over this lattice by considering any point to be connected to its close neighbors on the lattice, with weight determined by the aggregate posterior density ratio of that point to its neighbor.[3] This assigns a high cost to moving from a point with high density to a point with very low density, while prioritizing moving towards points with highest density. We then use Dijkstra's algorithm to find the shortest path on this graph between two query points. The output of this procedure is visualized in Fig. A5, along with reconstructions from the geodesic.

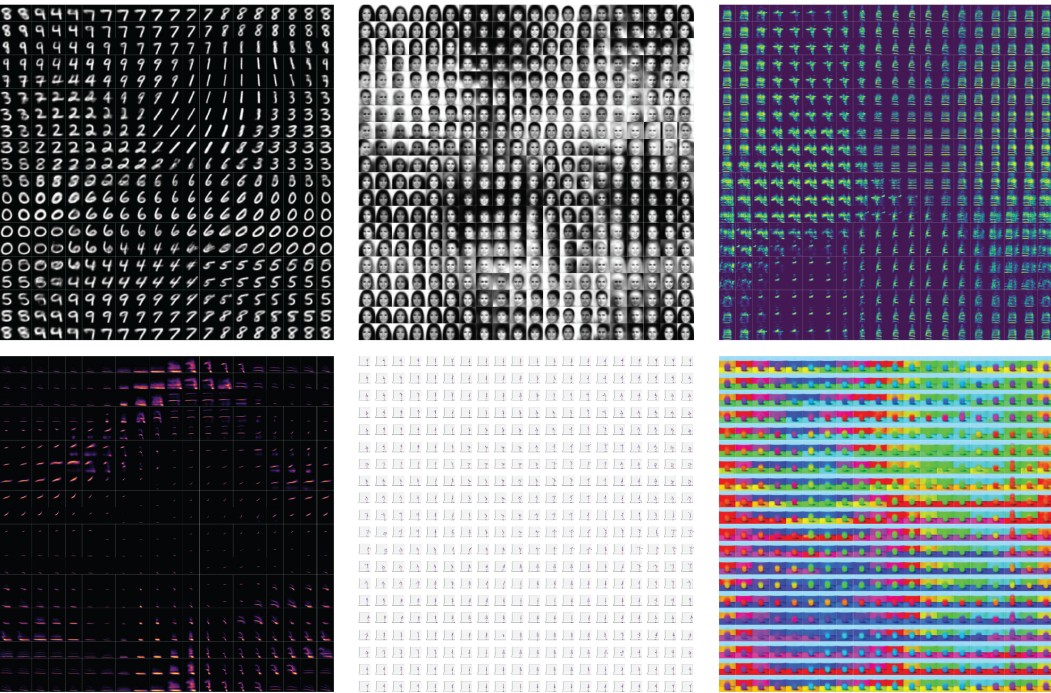

Figure A3: Traversals over the entire latent space for all datasets. Due to the constraints on our latent space (periodicity on the unit $d$-cube, 2-dimensional) we are able to fully explore the latent space for each model.

---

[3]Note that the density ratio defines one choice of a local metric, but others are also possible such as the pull back metric defined by the Fisher information matrix (Arvanitidis et al., 2018). We plan to pursue and compare these possibilities in future work.

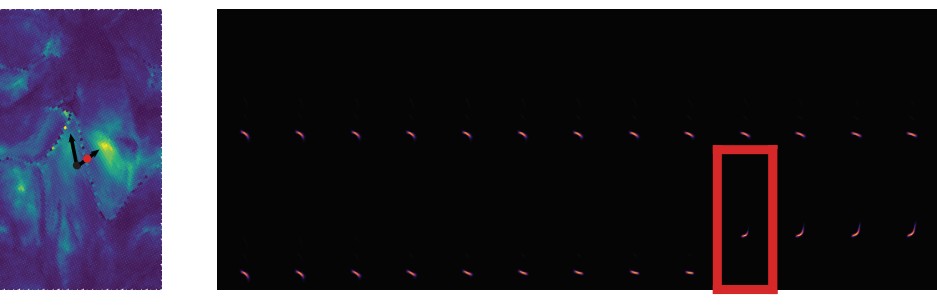

Figure A4: Latent traversal for the gerbil vocalization dataset. The top row is a traversal roughly parallel to what seems to be a class boundary; the bottom row crosses this boundary. The red dot and outlined red spectrogram correspond to the point at which this traversal crosses this boundary. This crossing is accompanied by a sharp change in the reconstruction.

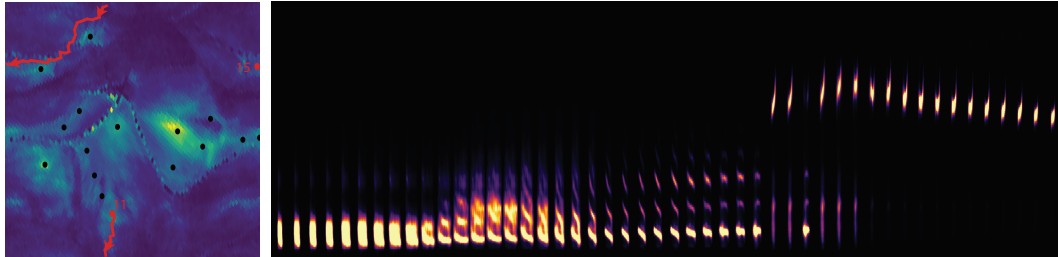

Figure A5: Geodesic from centroid 11 to centroid 15 (in Fig. 5B), and corresponding reconstructions

# D  **3dShapes** POSTERIORS AND JACOBIANS

Here, we visualize the aggregated posterior and squared Frobenius norm of the Jacobian for the **3dShapes** dataset analyzed in §3.4. The motivation for these analyses are described in §3.3, where they are demonstrated on **MNIST** and a dataset of gerbil vocalizations. Unlike in those datasets, we see no evidence of clustering—i.e. no peaks in the aggregated posterior or ridges in the scale of the Jacobian—in the QLVM embedding of the **3dShapes** dataset. Instead, we see a roughly uniform density over the entire latent space, reflecting the continuous nature of the generating latent factors for this dataset (Fig. A6).

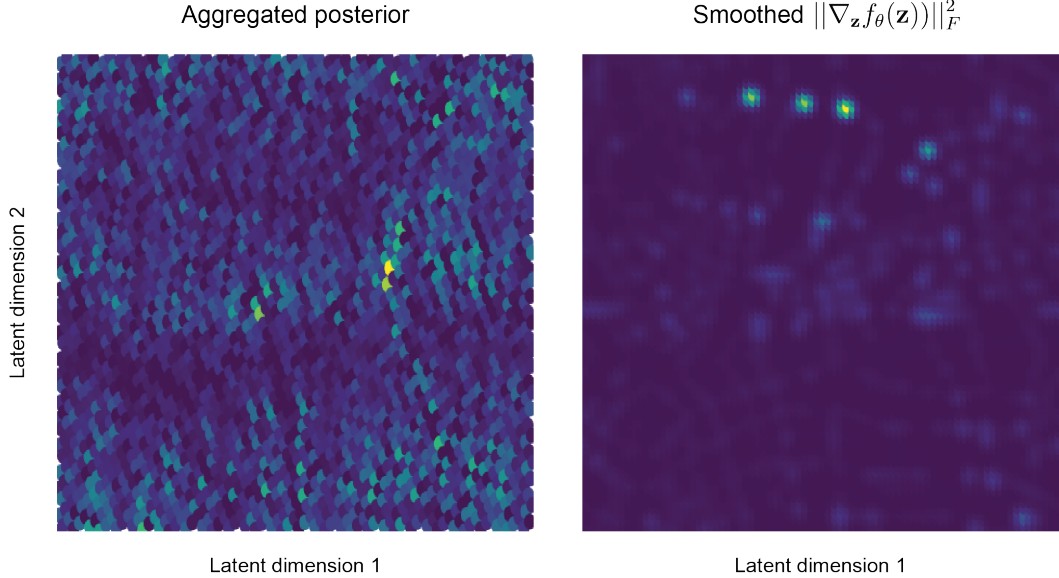

Figure A6: Aggregated posterior (left) and smoothed Frobenius norm of the decoder Jacobian (right). Unlike in MNIST and the gerbil vocal dataset, there is no visible clustering.

# E    MODEL ALTERATIONS

## E.1    MC,QMC,AND RQMC TRAINING

The bound of eq. (5) holds for any set of Monte Carlo samples — how important is the RQMC sampling scheme that we used? We trained networks with the same architecture and same number of latent samples per data point on **MNIST** using uniform MC samples and a fixed QMC lattice, finding that the RQMC sampling scheme that we used in the main text yielded small, but consistent improvements in performance. We hypothesize that this is because of the structure of the lattice combined with the uniform random shift; the uniform tiling of the latent space forces the model to explore the entire latent space every batch, while the shift ensures that the gaps in the latent space left by the lattice are explored across batches.

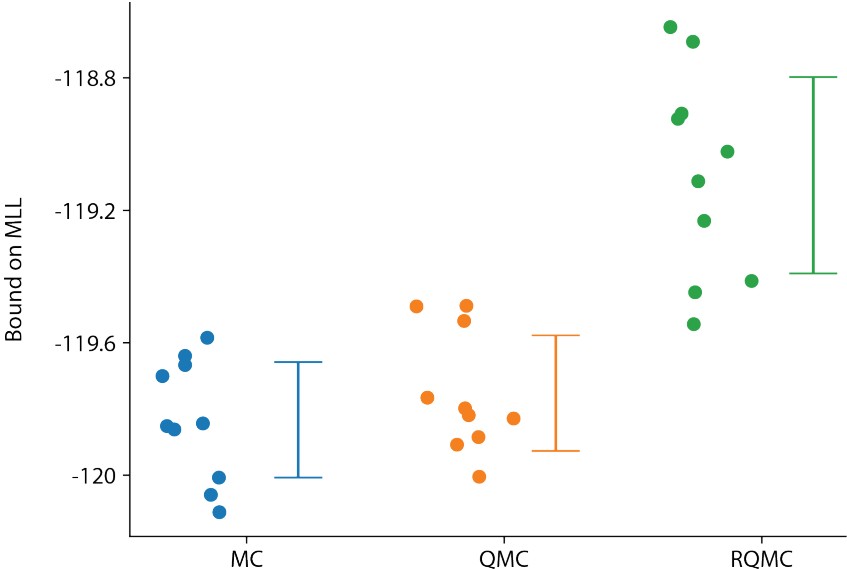

Figure A7: Comparison between model performance on **MNIST** test set using MC samples, a fixed QMC lattice, or RQMC (as in main text) for training. 10 models were trained per condition, RQMC consistently displays best performance.

## E.2    ALTERNATIVE LATENT SPACES

While lattices perform best when used for integration over periodic functions (see Owen, 2023, Chapter 16, section 7) , it may be the case that a flexible neural network decoder does not require periodicity (or can learn it implicitly) in order to learn a good representation for data. We demonstrate in Fig. A8 that this is not the case; rather, it impairs model performance (in terms of log evidence). As such, it appears that both enforcing periodicity is helpful for learning a good decoder.

Alternatively, rather than using a flat torus basis, we could use a Gaussian inverse CDF basis to effectively have a Gaussian prior over the latent space, while still relatively constraining the domain of the model. In Fig. A9, we compare performance of this basis to the QLVMs trained throughout the rest of the paper. This version performs similarly (but slightly worse) than QLVMs with a uniform prior and periodic boundary conditions on the latent space. We chalk up this minor difference to the fact that the Fibonacci lattice rule is not optimized for the Gaussian case. Regardless, the QLVM with a Gaussian prior outperforms VAE and IWAE models with the same prior.

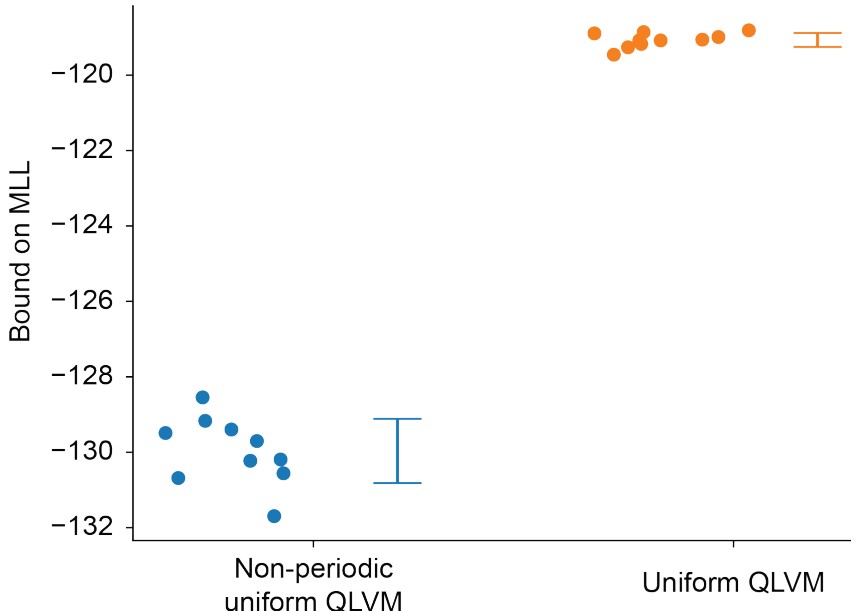

Figure A8: Comparison between model performance on **MNIST** test set using a non-periodic decoder, or a standard QLVM (as in the main text). 10 models were trained per condition. The standard QLVM consistently displays best performance.

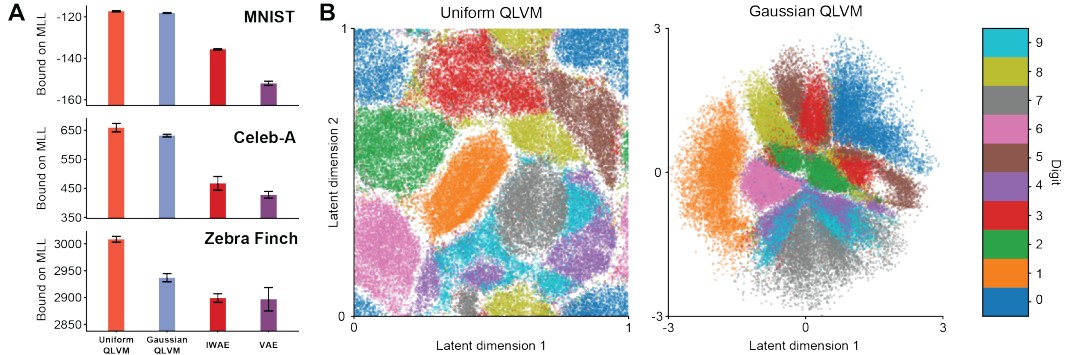

Figure A9: *(A)* Comparison between QLVMs trained using the default uniform prior, a Gaussian prior, 2D IWAEs, and 2D VAEs. In each case, both QLVMs outperform IWAEs and VAEs, with Gaussian QLVMs slightly underperforming uniform QLVMs. *(B)* Latent embeddings of `MNIST` the uniform QLVM and Gaussian QLVM.

## F    COMPARISON TO ITERATIVE AMORTIZED INFERENCE

An alternative approach to improving the performance of VAEs is to iteratively refine the variational distribution output by the encoder network, as proposed by Marino et al. (2018). While this strategy was shown to be effective for the higher-dimensional models, we demonstrate that it still underperforms 2D QLVMs (Fig. A10).

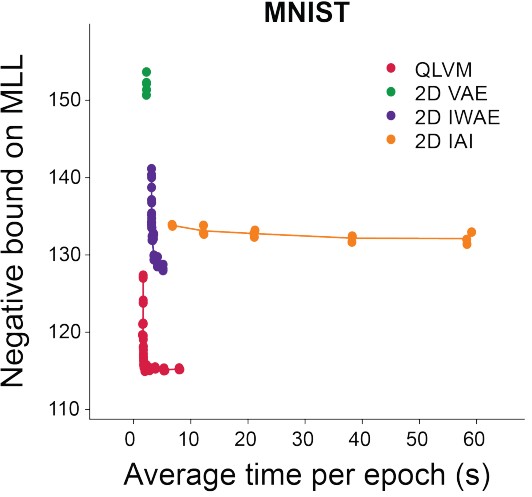

Figure A10: Performance vs. computational cost curves for 2D QLVMs (red), VAEs (green), IWAEs (purple), and iterative amortized inference-trained VAEs (orange) on `MNIST`. Curves closer to the lower left quadrant of each plot indicate a more favorable tradeoff.

## G    COMPARISON TO HIGHER-DIMENSIONAL VAEs

We additionally compared 2D QLVMs to VAEs and IWAEs of higher dimensions. Higher-dimensional IWAEs and VAEs outperformed 2D QLVMs quantitatively (Fig. A11); however, QLVMs qualitatively had similar reconstructions to up to 8D VAEs (Fig. A12). On simpler datasets, the improvements of higher-dimensional VAEs are relatively minor. However, samples from the QLVM prior both show higher sample quality and diversity than low-dimensional VAEs, and their samples more accurately reflect data than higher-dimensional VAEs (Fig. A13).

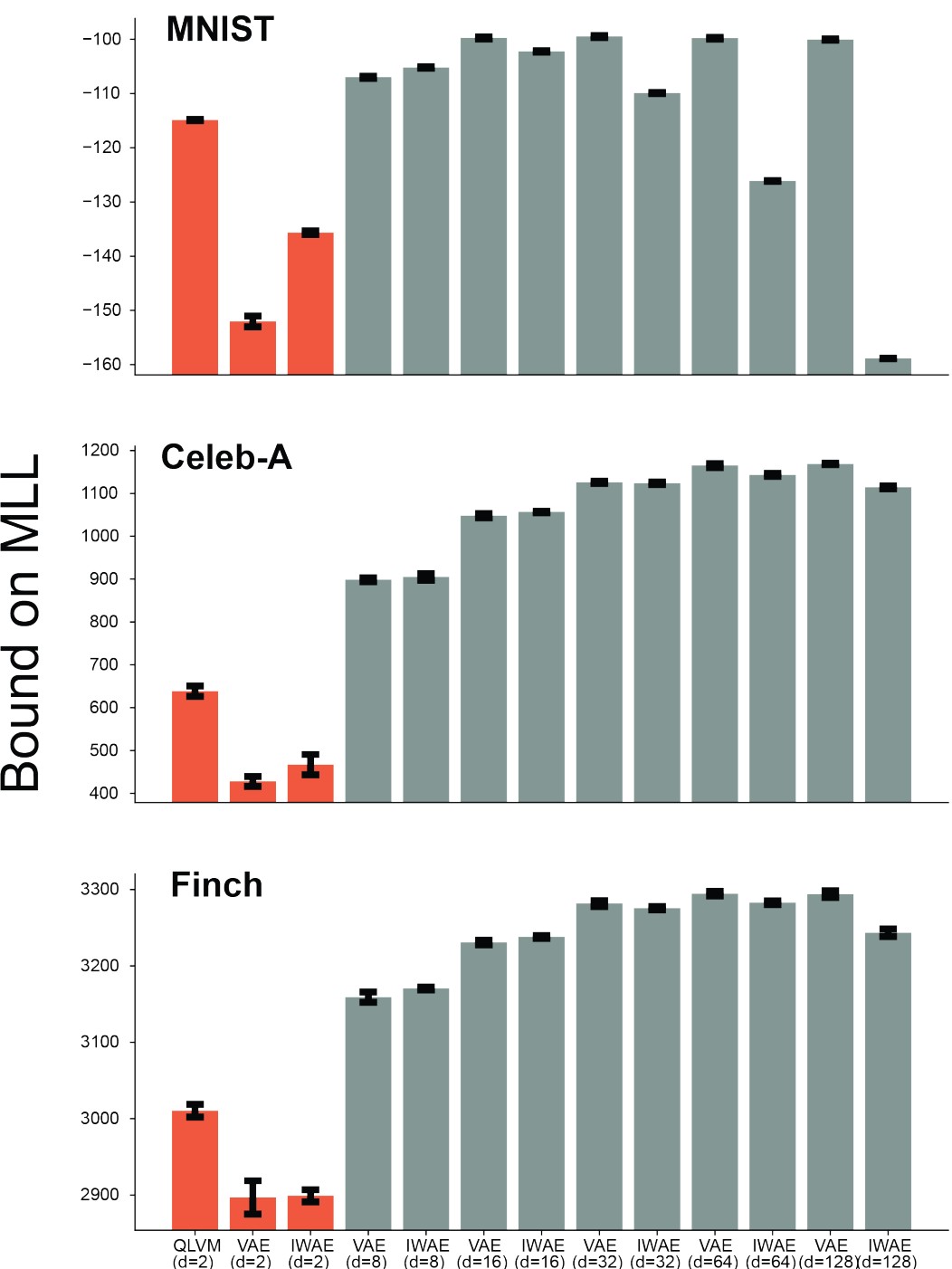

Figure A11: Bounds on marginal log likelihood on heldout test data for QLVMs, VAEs, and IWAEs of the same and larger dimensions; orange bars are 2D, gray bars are higher dimensional. Error bars indicate 1 standard deviation across five random seeds.

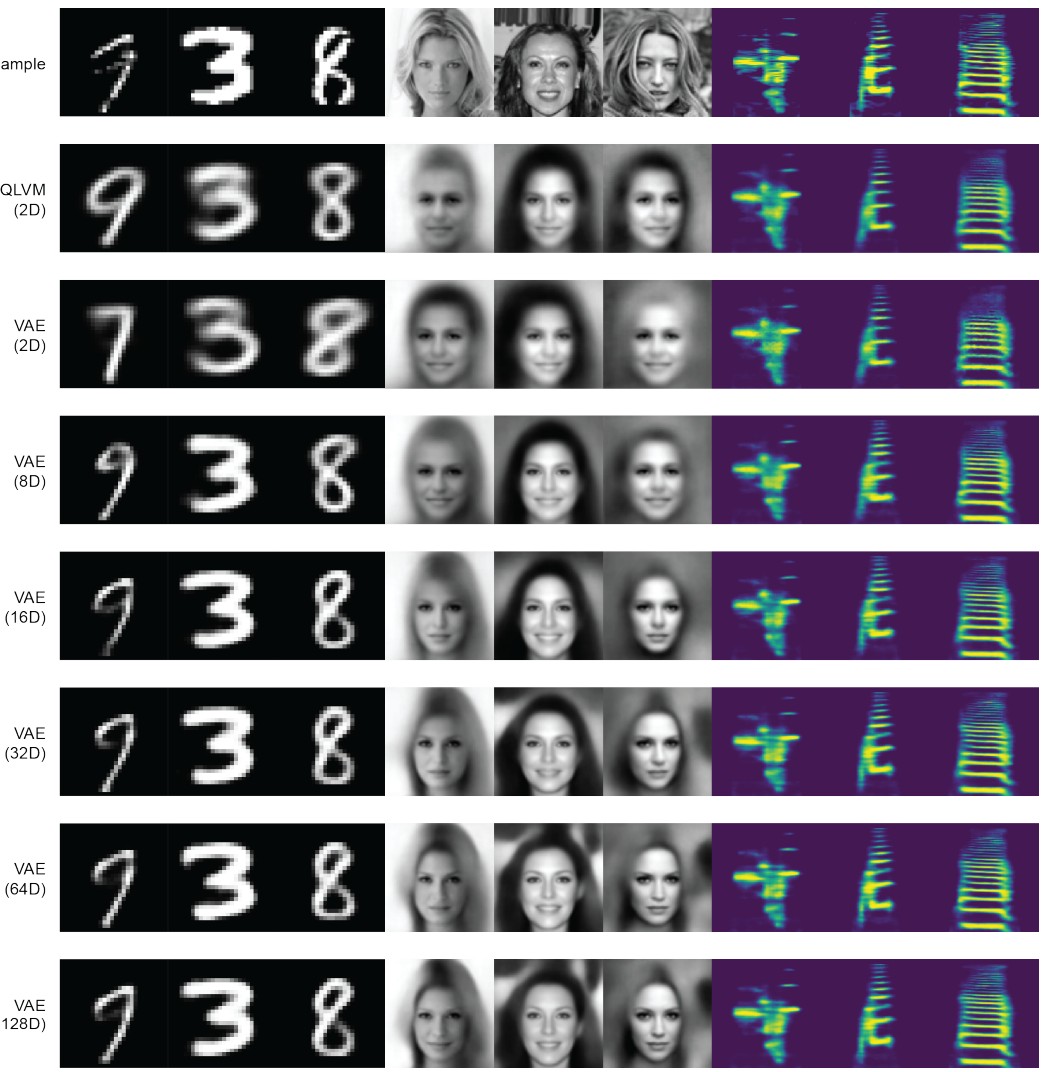

Figure A12: Reconstructions from models from Fig. A11; 2D QLVM reconstructions are comparable with up to 8D VAEs.

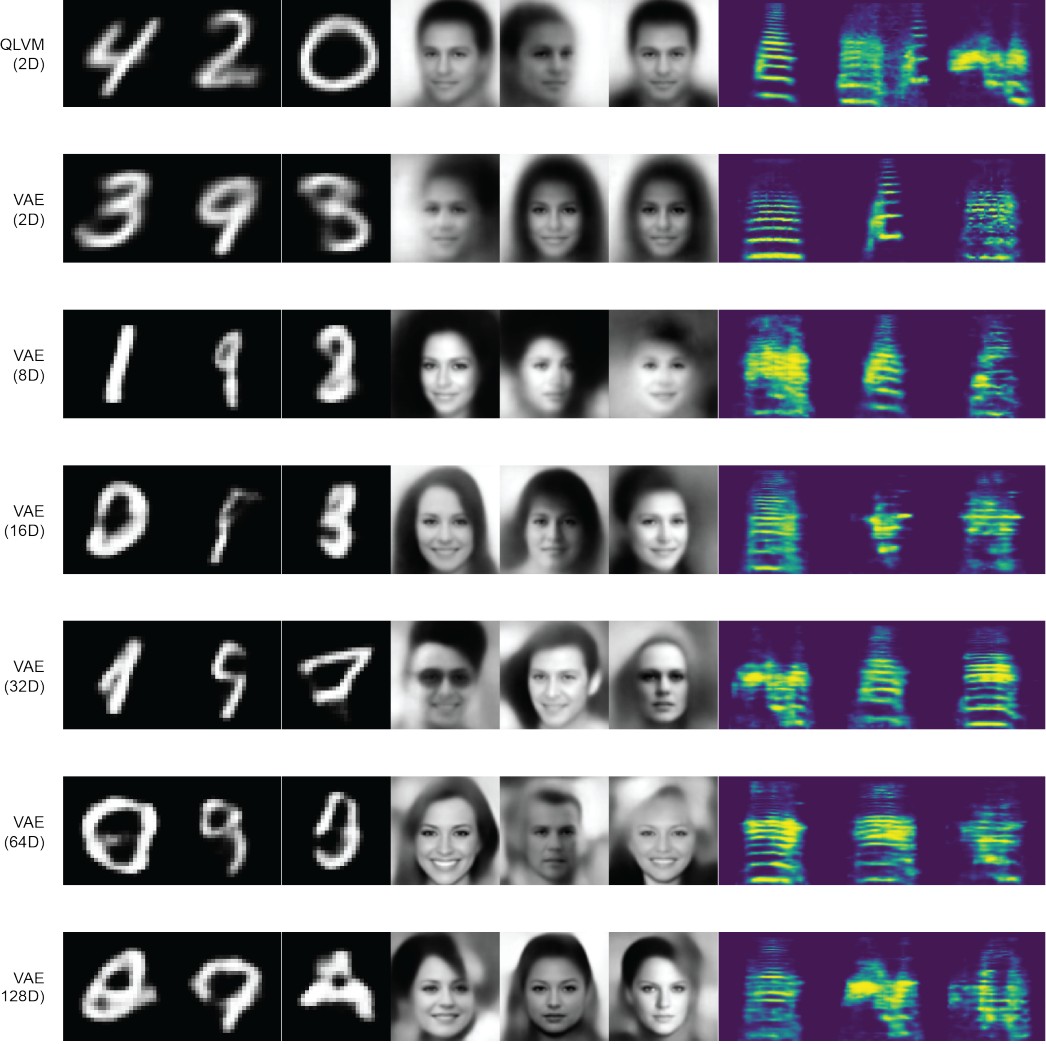

Figure A13: Samples from the prior for QLVMs and VAEs from Fig. A11; samples from QLVM prior resemble data more closely than samples from VAE priors.

