# OpenReview forum: "Quasi-Monte Carlo Methods Enable Extremely Low-Dimensional Deep Generative Models"
_ICLR.cc/2026/Conference — ICLR 2026 Poster_

### Official Review · Reviewer_YdPh · 2025-10-19

**Soundness:** 2
**Presentation:** 3
**Contribution:** 2
**Rating:** 2
**Confidence:** 5

**Summary:**

This paper investigates using explicit integration over the latent variable in deep latent variable models as an alternative to the amortized variational inference approach of VAEs. As this integral is intractable the authors propose using "quasi monte-carlo" integration; rathering than estimating via samples from the prior, the authors sample a randomly shifted lattice of points in the latent space to approximate the integral. The authors argue that this approach achieves considerably better results than variational auto encoders in cases where the latent space is low-dimensional enough that this approach is computationally feasible, i.e. 2 and 3 dimensions, cases where the latent embeddings may be used for visualization purposes. In their experiments, the authors show better performance compared to VAEs and importance-weighted auto encoders as measured by test loss and reconstruction. They also compare the approach to other low-dimensional embedding approaches tailored to visualization, such as UMAP.

**Strengths:**

**Writing**
- The paper is clear and well written with a nice description of the motivation of the problem.
- The paper includes a good discussion of related work and identifies many relevant papers from recent literature

**Methodology**
- Using randomly shifted lattices to tile the latent space for training is a clever idea and there does seem to be some notable performance improvements in some cases.
- There is a good exploration of the value of the lattice based approach in the appendix

**Results**
- The paper includes some excellect visualizations that make a compelling case for a deep-latent variable model with 2-dimensional uniform prior as a method for generating useful visualizations for high dimensional datasets.
- The authors used an interesting variety of datasets to showcase the performance of the approach

**Weaknesses:**

**Novelty**
- As the authors acknowledge, this idea is not inherently novel, though it has generally been dismissed as impractical in the past. I think that showing a compelling use case for this approach could be a worthwhile contribution, but it sets a high bar for the experimental results.

**Computational costs**
- This method has a high computational cost, even for the 2-d and 3-d cases discussed. The authors only test with relatively simple datasets and networks.
- The authors don't provide any quantitative comparison of the computational costs of different methods. This is a notable omission as computational concerns are the primary reason this approach hasn't been explored before.

**Framing**
- The authors frame this approach as generally better for low-dimensional latent variable models, but notably change the prior from the standard Gaussian used in most previous work to a uniform, without comparison their approach using a Gaussian prior.
- I think the uniform is a valid choice in this context for something like visualization, but it should be clear if this is not necessarily a general-purpose substitute for VAE-style methods.

**Baseline comparison**
- I'm not convinced that the comparison to baselines is fair.
- Looking at figure 2, the VAE seems to be suffering from posterior collapse. In my experience it's definitely possible to train a VAE for MNIST with 2-latent dimensions that performs better than what is shown.
- The IWAE baseline only used 10 samples, which seems quite small given the comparison to QLVMs. I would prefer to see a computation-time matched comparison to better understand if the QLVM approach is actually superior.
- The authors dismiss improved variational methods such as iterative amortized inference based on their computational expense and hyper parameters, but I would see these approaches as an important baseline. If they are truly less practical than this approach, I would like to see that justified experimentally.

**Experimental details**
- There are several details of the experiments that are not properly covered in the text as mentioned below.

**Questions:**

- Is the variational posterior approximation for the VAE and IWAE baselines still Gaussian in your experiments?
  - If so, how do you enforce the boundary constraints for the uniform prior and estimate the KL-divergence etc?
  - Does this affect the variance of the training objective e.g. compared to a Gaussian VAE where the KL-divergence can be computed in closed form?
- How is the test loss in figure 2 computed?
  - All three methods are computing different lower bounds on the MLL, so my question is: is this comparing learned decoder performance by evaluating all of the learned models with the same lower bound?
  - Or alternatively is it reporting the same loss used to train each model?

---

> ### Author Response · Authors · 2025-11-20
> **Rebuttal**
>
> Thank you for your thoughtful comments. We are optimistic that we can address all of your concerns.
>
> > As the authors acknowledge, this idea is not inherently novel
>
> While simple Monte Carlo has been mentioned off hand in prior work, it hasn’t received serious consideration. We therefore feel the method itself *is novel* in the sense that nobody has published an explicit implementation. We also describe nontrivial refinements such as using QMC and sharing samples across elements in a mini-batch (a benefit over IWAEs that we under-emphasized in the first version of the paper).
>
> Furthermore, very few papers have highlighted the empirical benefits of 2D latent spaces—e.g. visualizing the Jacobian of the decoder to aid clustering. Thus, we hope that our empirical demonstrations can also be taken into account as novel contributions.
>
> > This method has a high computational cost, even for the 2-d and 3-d cases discussed.
>
> In an effort to be transparent about limitations, we were unclear. While there is some computational cost to our approach, it is far from prohibitive. Based on your suggestion, we now explicitly characterize the performance vs. computation tradeoff in a new section and figure (sec. 3.2 and Fig. 4 in the revision). We show that QLVMs outperform VAEs and IWAEs even when computational expenses are matched.
>
> > The authors … change the prior from the standard Gaussian used in most previous work to a uniform … it should be clear if this is not necessarily a general-purpose substitute for VAE-style methods.
>
> This is a good point. We neglected to mention that it is extremely easy to change the prior distribution for the QLVM by mapping the latent space through a fixed nonlinearity given by the inverse CDF of the prior (see [Art Owen’s book](https://artowen.su.domains/mc/) Chpt 4).
>
> In Section 2.2 of the revision, we added a paragraph on “Changing the prior distribution” to explain how easy this is. Further, we have added Supp. Fig 9 to show a QLVM with a Gaussian prior. It has similar performance to the QLVMs with a uniform prior, and it still outperforms VAE and IWAE baselines.
>
> > In my experience it's possible to train a VAE for MNIST with 2-latent dimensions that performs better than what is shown.
>
> Thank you for prompting us to revisit our VAE and IWAE implementations. After doing more systematic sweeps over hyperparameters, we were able to improve these baselines and we’ve updated all of our figures accordingly. Despite this, the both models still underperform the QLVM (Fig. 2), and variational posteriors do not match empirical posteriors on 2 out of 3 datasets (Fig. 3). We consulted various open-source implementations of VAEs (such as [this one](https://github.com/lyeoni/pytorch-mnist-VAE)) to ensure correctness.
>
> We think that it is important to note that *we hardly did any hyperparameter tuning for the QLVM models* other than to pick a somewhat reasonable learning rate that doesn’t explode. This “ease-of-use” is not easy to quantitatively benchmark, but we hope it can be taken into account.
>
> > The IWAE baseline only used 10 samples, which seems quite small given the comparison to QLVMs. I would prefer to see a computation-time matched comparison…
>
> We have now included this comparison by adding Fig 4. QLVMs still outperform IWAEs across a variety of sample sizes. [Rainforth et al. (2018)](https://proceedings.mlr.press/v80/rainforth18b.html) provide a compelling justification for why increasing sample size doesn’t always improve IWAE performance.
>
> Our first submission did not emphasize an important fact about this comparison. In the QLVM, the same latent samples can be re-used across a minibatch of images because they are drawn from the same prior distribution. This is not the case for the IWAE, where each element of the minibatch has its own proposal distribution given by the encoder network. Thus, following our notation in section 2, the number of latent samples used by the IWAE is actually `m * batch_size` while the number of latent samples used by the QLVM is `m`. This explains why we were able to use many more samples for a QLVM vs. an IWAE.
>
> > The authors dismiss… iterative amortized inference based on their computational expense and hyper parameters … I would like to see that justified
>
> We were able to include a comparison of QLVMs to [iterative amortized inference](https://github.com/joelouismarino/iterative_inference) on MNIST in the revision (see Appendix F, Figure A10). Other than reducing the latent space from 64D to 2D, we keep their parameters as reported in their appendix C.7. As expected, this baseline is 10x slower than the QLVMs used in the main text (13s per epoch as opposed to 1-2s). Even when using 5 iterations per batch, their approach is slower than a QLVM using an extremely dense lattice (~50,000 points), and the performance is similar to the IWAE baseline on MNIST. If you think it would be helpful, we could incorporate this comparison into Fig. 4 of the main text.

---

> > ### Comment · Reviewer_YdPh · 2025-11-26
> >
> > Thank you to the authors for their very thorough rebuttal and answers! These largely addressed my concerns about the paper and I will update my score to recommend acceptance.
> >
> > Regardless of the outcome, I would strongly recommend the authors to continue their empirical study of this approach and to consider releasing a refined codebase. It would be useful to see a wider comparison to a greater variety of datasets such as others addressed in the UMAP paper along with practical recommendations for how to design the decoder and how to apply it to different types of data.

---

> > > ### Author Response · Authors · 2025-12-02
> > >
> > > Thank you very much for this discussion! We appreciate the time and effort you put into your review. We are excited to continue testing this approach, and will work towards a refined codebase that others can use and explore as well.

---

> ### Author Response · Authors · 2025-11-20
> **Responses to Questions**
>
> Please find responses to your two questions below. Thanks again for your feedback and review of our work.
>
> > Is the variational posterior approximation for the VAE and IWAE baselines still Gaussian in your experiments?
>
> The prior and variational posterior approximation for VAE and IWAE baselines are still Gaussian; we have added text (lines 801-802) to clarify. It is also possible to train QLVMs with respect to a Gaussian prior (Supp. Fig 9) and this variant has comparable performance to the QLVM with the uniform prior &mdash; i.e. still outperforms VAE and IWAE baselines.
>
> > How is the test loss in figure 2 computed?
>
> The loss reported in this figure is the QMC estimate for QLVMs and the ELBO for IWAEs and VAEs on the test set. We have added text (line 228) to clarify. In figure 3 we additionally compare a QMC estimate of the marginal log likelihood defined by the VAE and IWAE decoders; we find that QLVMs still outperform (though applying QMC to estimate the marginal log likelihood in a VAE or IWAE does improve the bound on these baselines, indicating imperfections in the encoder networks).

---

### Official Review · Reviewer_Q9q7 · 2025-10-31

**Soundness:** 3
**Presentation:** 3
**Contribution:** 2
**Rating:** 6
**Confidence:** 4

**Summary:**

This paper takes an interesting approach to latent variable modelling. The presentation contrasts the approximations done using amortized inference with a more direct inference by using a monto carlo estimate of the marginal log-likelihood. The approximation to the marginal log-likelihood is done using uniform prior which means that the latent space needs to be bounded and for computational reasons very low-dimensional in order to get a good estimate. The paper discusses the pros and cons of using this more "natural" approach compared to an amortized inference mechanism used in VAEs.

**Strengths:**

This is well written paper that is very easy to understand. In some sense I think the narrative of explaining this and continously relating this to VAEs is a little bit too specific, in some ways this is a much more natural approach to the problem of a latent variable models compared to an amortized inference scheme. That is just my personal oppinion and not something that has been reflected in the score. The flip side of this is that the latent space is required to be very low-dimensional in order to be able to compute the approximation. As such, and this the authors do very well, it shouldn't really be put in the same context as VAEs or other generative models but rather UMAP t-SNE and possible a GP-LVM as its where these models are useful this approach will be applicable. Differently from the first two though, this is a generative model and allows for the latent location to be infererred for new data, due to the uniform latent prior, which is more challenging with spectral dimensionality reduction methods.

I liked the discussion around the failure modes of the amortized approximate posterior. I think your intuitions about this makes sense and it would be nice to evaluate this a bit more to get a more. There is a very nice paper [1] that discusses the interplay of the complexity of the back and the generative mapping in an auto-encoder setting. There might be some information that is relevant there for future work.

[1] Barber, D. (2014). Implicit Representation Networks.

**Weaknesses:**

While I think the paper is a nice study and tells a story about how one can approach latent variable modelling in a more direct way there is also a question about the limitations of the paper. The first thing I think would have been interesting to see is the implication of the uniform prior over the latent represenation and what effects it does have. It would be great to have a discussion on more informative priors and how the approach would change in that case and the impact on the results. I don't fully understand the results right at the end of the Appendix which I assume are supposed to relate to this.

The authors state and are very honest with the fact that the approach is very computationally expensive. It would be nice to see what this actually means in-terms of numbers and how many samples \(M\) that is used in the experiments and the dependency of this.

**Questions:**

Given that you are able to compute the Jacobian, can't you then represent the pull back metric in the latent space? This would allow you to compute more principled Geodesics compared to the approach you are now taking.

---

> ### Author Response · Authors · 2025-11-20
>
> Thank you for your thoughtful comments and positive summary of our work. We address your main concerns below.
>
> > I think would have been interesting to see is the implication of the uniform prior over the latent represenation and what effects it does have. It would be great to have a discussion on more informative priors and how the approach would change in that case and the impact on the results.
>
> Great point, Reviewer YdPh raised a similar comment.
>
> We neglected to mention this, but it is very easy to change the prior distribution for the QLVM by mapping the latent space through a fixed nonlinearity given by the inverse CDF of the prior (see, e.g., [Art Owen’s book](https://artowen.su.domains/mc/) Chpt 4). We’ve rectified this in the revision by adding a paragraph to the bottom of section 2.2 and adding Figure A9 which shows a QLVM with a Gaussian prior.
>
> Perhaps unsurprisingly, the performance of the QLVM is not very sensitive to changing the prior. This is consistent with the work of [Hyvarinen & Pajunen (1999)](https://doi.org/10.1016/S0893-6080(98)00140-3), who showed that the prior distribution is not unique or identifiable in nonlinear latent variable models.
>
> While the choice of prior does not seriously impact the reconstruction error, Supp Fig 9 shows that it does heavily influence the appearance of the low-dimensional visualization of the dataset. In this respect, the uniform distribution is much nicer as the embedded points become more evenly distributed over the latent space.
>
> > The authors state and are very honest with the fact that the approach is very computationally expensive. It would be nice to see what this actually means in-terms of numbers and how many samples (M) that is used in the experiments and the dependency of this.
>
> In an effort to be transparent, we think we inadvertently overstated these computational limitations of our approach. Your suggestion to explicitly characterize the tradeoff between performance and computational costs is an excellent one, and we’ve now added this as a new section in the revision (section 3.2 and Figure 4). We hope this clarifies that the computational costs of the QLVM are roughly comparable to VAE/IWAE baselines for 2D latent spaces.

---

### Official Review · Reviewer_6SFt · 2025-10-31

**Soundness:** 4
**Presentation:** 3
**Contribution:** 4
**Rating:** 8
**Confidence:** 5

**Summary:**

This paper investigates the performance of QMC in computing the marginal log likelihood in very low-dimensional settings. This resulted in a model that, although computationally demanding, did not require an encoder and outperformed VAEs and IWAEs on simple datasets, such as MNIST. Given the dimensionality limitation of such models, the authors propose this method as an alternative to UMAP or t-SNE. In opposition to these algorithms, QLVMs benefit from additional capabilities, such as latent traversal or density estimation, which are essential for unsupervised data analysis, ensuring transparent results.

**Strengths:**

This paper investigates a simple yet overlooked way of computing the marginal likelihood and proposes a creative approach to use this in the restricted setting of 3 or fewer latent dimensions. I believe that this paper can have a strong impact on the unsupervised learning community, especially for people doing exploratory data analysis on high-dimensional data. The paper was easy to follow, and I really enjoyed reading it. Well done to the authors, great work!

**Weaknesses:**

- The posterior collapse issue of VAEs when reducing the number of latent dimensions too much has been overlooked. It would be great to add this to the discussion in 3.1.
- Fig. 3 B do not explain which models are associated with each line.

**Questions:**

- In 3.1, VAEs in very low-dimensional spaces may suffer from posterior collapse [1-3], it could easily be checked (e.g., following the method proposed in [3] to monitor passive variables). Such results would provide a good explanation for the poor performances of VAEs in this setting. I am not sure whether IWAE can also suffer from posterior collapse, but this could be worth exploring.
- Also in 3.1, VAEs tend to behave in a polarised regime [2-4] where some latent variables are used to decrease the KLD while others encode meaningful information but depart from the prior. Could what is observed in Fig. 3 be a consequence of this?
- In Fig. 3 B, which line corresponds to which model?
- Have the authors tried QLVM on high-dimensional tabular data (e.g., gene expression)? Which type of results would they expect?
- l. 123 it should be ")]" instead of "])"
_________

**References**

[1] Dai, B., Wang, Z., & Wipf, D. (2020, November). The usual suspects? Reassessing blame for VAE posterior collapse. In International conference on machine learning (pp. 2313-2322). PMLR.

[2] Rolinek, M., Zietlow, D., & Martius, G. (2019). Variational autoencoders pursue PCA directions (by accident). In Proceedings of the IEEE/CVF Conference on Computer Vision and Pattern Recognition (pp. 12406-12415).

[3] Bonheme, L., & Grzes, M. (2023). Be more active! understanding the differences between mean and sampled representations of variational autoencoders. Journal of Machine Learning Research, 24(324), 1-30.

[4] Dai, B., Wang, Y., Aston, J., Hua, G., & Wipf, D. (2018). Connections with robust PCA and the role of emergent sparsity in variational autoencoder models. Journal of Machine Learning Research, 19(41), 1-42.

---

> ### Author Response · Authors · 2025-11-20
>
> Thanks so much for your positive comments and review of our work.
>
> > The posterior collapse issue of VAEs when reducing the number of latent dimensions too much has been overlooked. It would be great to add this to the discussion in 3.1.
>
> Thanks for this suggestion. In response to concerns from reviewer YdPh we revisited our VAE and IWAE implementations and have revised both the figure and text in this section of the paper.
>
> Our revision adds a more rigorous statistical test showing that VAE and IWAE bounds are consistently loose. However, we don’t think the problem is posterior collapse, exactly. We observe that the variational posteriors output by the encoder networks are very, very low variance. For the ELBO, this indicates to us that the likelihood term is dominating over the KL divergence term to the prior. This is in some sense the opposite failure mode to posterior collapse, where the model is insensitive to the likelihood and only tries to fit the prior. Let us know if you think that interpretation makes sense and whether it comes across in the revision.
>
> In the meantime we will also take a closer look at the citations you included in your review to see if we can find a deeper connection with our results.
>
> > Fig. 3 B do not explain which models are associated with each line.
>
> Thanks for pointing this out. We have updated Figure 3 in the revision. Please let us know if anything is still unclear.
>
> > Also in 3.1, VAEs tend to behave in a polarised regime [2-4] where some latent variables are used to decrease the KLD while others encode meaningful information but depart from the prior. Could what is observed in Fig. 3 be a consequence of this?
>
> It seems to us that in the very low-dimensional setting the VAEs are highly incentivised to use both dimensions and therefore depart strongly from the prior along both axes. However, as we add more latent dimensions to the VAE models we do believe we observe what you’re seeing. It would be an interesting follow-up study to explore how the VAE solution changes as a function of the latent dimensionality.
>
> > Have the authors tried QLVM on high-dimensional tabular data (e.g., gene expression)? Which type of results would they expect?
>
> This is an excellent suggestion, and one that we have discussed exploring in future work. We would be excited to apply QLVMs to this data type.

---

> > ### Comment · Reviewer_6SFt · 2025-11-26
> > **Thank you for the answer**
> >
> > Thanks for the interesting discussion. Fig. 3 is clear to me now.
> >
> > Regarding the discussion on posterior collapse:
> >
> > >  For the ELBO, this indicates to us that the likelihood term is dominating over the KL divergence term to the prior. This is in some sense the opposite failure mode to posterior collapse, where the model is insensitive to the likelihood and only tries to fit the prior. Let us know if you think that interpretation makes sense and whether it comes across in the revision.
> >
> > In the polarised regime, you have two types of variables, active and passive. The active variables will encode useful information for the reconstruction and diverge from the prior. Specifically, an active variable will have a low variance $\sigma \approx 0$, and thus during reparametrization $z = \mu + \sigma \epsilon \approx \mu$. Passive variables are doing the opposite, not encoding anything useful but lowering the KL divergence such that $\mu \approx 0$, $\sigma \approx 1$, and during reparametrization $z = \mu + \sigma \epsilon \approx \epsilon$ so $z \sim \mathcal{N}(0, 1)$.
> >
> > Looking at Fig. 3 B you are right in saying that this is not a case of posterior collapse. It looks like your VAE has indeed 2 active variables because for both latent variables it seems that $\sigma \approx 0$. This probably explains why the bound is loose, given that the latents strongly diverge from the prior here. Reconstructing well is likely more beneficial than being close to the prior for the model, this would probably change by weighting the KLD with $\beta > 1$ as in [1].
> >
> > [1]  Higgins, Irina, et al. "beta-vae: Learning basic visual concepts with a constrained variational framework." International conference on learning representations. 2017.

---

> ### Author Response · Authors · 2025-11-26
> **Follow up**
>
> Thank you for this discussion. It is interesting to think about a comparison to a $\beta$-VAE, as you mention. We will train some networks with $\beta > 1$ as well as some networks with $0 \leq \beta < 1$ and see what comes out. Although these models are no longer optimizing an (approximate) lower bound on the evidence, it would be interesting to see how they behave in the very low-dimensional setting we're interested in. We suspect that when $\beta = 0$ the results will be comparable to what we observed when $\beta = 1$. This would confirm our intuition that VAEs indeed behave like classical autoencoders in the very low-dimensional regime.
>
> We agree with your prediction that when $\beta$ is sufficiently larger than one we will see a regime where one of the two latent dimensions is collapsed (the "polarised regime") and a regime where both dimensions are fully collapsed. We will run these experiments and consider adding this as an additional supplemental result to the appendix. Thanks again for the suggestion.

---

### Official Review · Reviewer_2qje · 2025-11-09

**Soundness:** 3
**Presentation:** 3
**Contribution:** 3
**Rating:** 6
**Confidence:** 4

**Summary:**

The authors propose a latent variable model that directly optimizes the marginal likelihood using Monte Carlo integration, without using an encoder-based approximate posterior. For uniform priors, the authors provide appropriate integration schemes when the latent space has small dimensionality. The experimental results show that the proposed approach is competitive with related deep generative models, and the learned latent representations appear meaningful.

**Strengths:**

- The proposed method is conceptually simple, straightforward to implement, and performs well in the conducted experiments.
- The integration schemes, together with the uniform prior, appear to leverage the latent space effectively.
- The paper is generally well written and easy to follow.
- I think that the method provides a meaningful alternative to UMAP-style approaches, with the advantage that it includes a decoder.

**Weaknesses:**

- As acknowledged by the authors, scalability is a clear limitation of the approach.
- While the experimental results are promising, it is not completely clear to what extent this approach opens new research questions beyond the presented scenarios.

**Questions:**

Q1. The structure of the latent space is somewhat unclear due to the periodic boundary. Could the authors elaborate on the choice $\mathbf{z} = (\sin \mathbf{z}, \cos \mathbf{z})$ and its implications? What is the topological structure of the resulting latent space?

Q2. In Fig. 1, it appears that blue points are always associated with batch 1 and red points with batch 2. Are the batches fixed throughout training, or do they change? In other words, do data points in batch 1 always interact with the same set of latent points?

---

> ### Author Response · Authors · 2025-11-20
>
> We appreciate your thoughtful review of our work.
>
> > As acknowledged by the authors, scalability is a clear limitation of the approach.
>
> This point was consistently raised by all of the reviewers, leading us to believe that we overstated the computational limitations of our approach (in an effort to be transparent about them!). We now directly address the computational cost in section 3.2, showing that for 2D latent spaces the computational cost is comparable between QLVMs and VAEs/IWAEs. Even resource-constrained QLVMs with fewer lattice points tend to outperform VAEs + IWAEs.
>
> Note that scalability to higher dimensional latent spaces is still a limitation. Our new results establish that the computational cost of training a 2D QLVM model is comparable to fitting a VAE or IWAE over a 2D latent space. However, given the broad popularity of methods like UMAP, we think that learning 2D latent spaces is useful.
>
> > While the experimental results are promising, it is not completely clear to what extent this approach opens new research questions beyond the presented scenarios.
>
> We are currently using the QLVM to dig deeper into the gerbil vocalization dataset shown in Fig 4 of the initial submission (Fig 5 in the revision). The model reveals very rich, and scientifically interesting, details. Unfortunately, it is difficult to fit these into the current manuscript&mdash;a proper report will require us to write a follow-up paper. We hope that the reviewer will be sympathetic to this perspective and accept the clustering results and comparisons to UMAP in Fig. 6 as a sufficient proof-of-concept.
>
> Moreover, we think there are several extensions to the QLVM which are of immediate interest to the latent variable modeling community. We have added a new section to the discussion on possible model extensions to flesh out these ideas (see bottom of page 10).
>
> > The structure of the latent space is somewhat unclear due to the periodic boundary. Could the authors elaborate on the choice and its implications? What is the topological structure of the resulting latent space?
>
> Certainly. The topological structure is a flat torus (i.e. a torus with equal curvature everywhere). One could remove the periodic boundary conditions without changing the outcome very much.  The motivation for using periodic boundaries in 2D is that one can prove that the Fibonacci lattice is the best way to cover the flat torus with a discrete set of points. With finite nonperiodic boundaries, it is difficult to find optimal point configurations (see sphere packing problems).
>
> We also found that the periodic boundaries are nice to prevent islands or clumps of points ending up “stuck” in the corners of the 2D latent space. Thus, the uniform prior combined with periodic boundary conditions helps spread the embedded points out uniformly and neatly for visualization. We also do see some (marginal) improvement when fitting a model with periodic boundary conditions. We have added brief remarks about this in Appendix E2 and Figure A9.
>
> > In Fig. 1, it appears that blue points are always associated with batch 1 and red points with batch 2. Are the batches fixed throughout training, or do they change? In other words, do data points in batch 1 always interact with the same set of latent points?
>
> The points in Fig. 1 are examples of possible points; each batch, we sample a random shift of a fixed lattice (modulo 1), so data points in each batch interact with different sets of latent points throughout training. We have added a paragraph in section 2.3 to clarify this point (see “Computational Benefits of QLVMs”). When gradients are evaluated using minibatches, we share latent points across all samples in the batch. This is a subtle, but important computational advantage of the QLVM over the IWAE. In the latter, each element of the minibatch interacts with a separate set of sampled latents. Thus, for a GPU memory budget that allows `M` latent samples per minibatch, the IWAE can only use `M / batch size` latent points per image. The QLVM is able to share the latent samples across all images in the minibatch, because the samples are drawn from the same prior distribution.

---

### Author Response · Authors · 2025-12-03

We want to thank the ACs for dealing with the challenging situation brought on by the leak of reviewer + AC identities.

Before the discussion period was cut short, we had a productive and amicable discussion with Reviewer YdPh who had asked for additional benchmark experiments. We want to draw your attention to their public comment on Nov 26, *"Thank you to the authors for their very thorough rebuttal and answers! These largely addressed my concerns about the paper and I will update my score to recommend acceptance."*

The other three reviewers gave strong initial scores to our paper. All four reviewers gave us great feedback which strengthened the manuscript. We are particularly happy that we were able to include a new Figure 4 (performance vs. computational cost curves), which was spurred by several reviewer comments and which we think was critical for convincing Reviewer YdPh.

Please don't hesitate to inquire if there is more information or context that we can provide. Thank you again for the extra work you are putting into meta-reviews this year.

---

### Meta-Review · Area_Chair_6i62 · 2026-01-06

**Summary:**

The paper proposes Quasi-Monte Carlo Latent Variable Models (QLVMs), a deep generative framework specialized for extremely low-dimensional latent spaces ($d \le 3$). By replacing the standard variational encoder of VAEs with a direct, brute-force approximation of the marginal likelihood using Randomized Quasi-Monte Carlo (RQMC) integration on a lattice, the authors aim to produce superior, interpretable embeddings. This approach positions itself as a generative alternative to visualization techniques like UMAP or t-SNE, offering features like density estimation and geodesic traversals.

Upon reviewing the submission and initial feedback, the primary obstacles to acceptance were the computational feasibility of the method and the rigor of the comparative evaluation. Specifically, relying on brute-force integration raises legitimate scalability concerns, making it crucial to determine if the method offers a real advantage over VAEs/IWAEs under a fixed computational budget. Furthermore, there were valid questions regarding whether the baseline VAEs were suffering from posterior collapse (artificially inflating the proposed method's relative performance) and if the specific choice of uniform priors limited the method's generalizability compared to standard Gaussian assumptions.

I recommend acceptance. The authors provided a very effective rebuttal that resolved these concerns. They introduced a Pareto front analysis (Figure 4) proving that QLVMs outperform VAEs and IWAEs even when matched for wall-clock time, effectively settling the efficiency debate for low-dimensional settings. Additionally, they addressed the theoretical questions regarding the "flat torus" topology and demonstrated the method's flexibility with Gaussian priors (Appendix E.2). Given that the dissenting reviewer explicitly reversed their stance to recommend acceptance based on these new results, the paper now stands as a robust contribution to representation learning for visualization.

**Reviewer Concerns:**

The **addressed concerns** raised are as follows:
* **Computational Cost & Performance (Reviewer YdPh, Q9q7): Addressed by the new Section 3.2 and Figure 4. The authors showed that while the method doesn't scale to high $d$, it is efficient enough in low $d$ to beat baselines on a fixed compute budget.
* Baselines & Fairness (Reviewer YdPh): Addressed by retuning VAE/IWAE baselines and adding an "Iterative Amortised Inference" comparison (Appendix F), which showed the QLVM still superior in this regime.
* Topology & Priors (Reviewer 2qje, Q9q7): The authors clarified the "flat torus" topology induced by periodic boundaries and provided evidence (Appendix E.2) that the method works with Gaussian priors via inverse CDF mapping.
* Theoretical Interpretation (Reviewer 6SFt): The discussion clarified that VAEs in this regime suffer from a "polarised" regime (active/passive variables) rather than simple posterior collapse.

The **outstanding concerns** raised, although fewer, are as follows:
* Scalability: The method is strictly limited to low dimensions ($d \le 3$). All reviewers accepted this as a boundary condition of the paper, given its focus on interpretable visualisation.
* Geodesics: Reviewer Q9q7 suggested using a pullback metric for geodesics. The authors acknowledged this as future work.

**Reviewer Scores:**

* **Reviewer YdPh. Original score 2. Estimated score: 6**. In the discussion, this reviewer explicitly stated: "These largely addressed my concerns about the paper and I will update my score to recommend acceptance." Given the strength of the rebuttal, I estimate a jump to an 6 and potentially to even 8.
* **Reviewer 6SFt. Original score 8. Estimated score: 8**. They were highly enthusiastic ("excellent work") and satisfied with the clarifications on Fig 3B and posterior theory. Score likely remains unchanged.
* **Reviewer 2qje. Original score 6. Estimated score: 6**. They requested an extension and clarification on lattice topology (flat torus) and batching. The authors answered clearly. The reviewer likely maintained their positive, albeit cautious, score.
* **Reviewer Q9q7. Original score 6. Estimated score: 6**. Their concerns about priors and computational costs were addressed with new figures. They are likely a solid 6 or potentially an 8.

---

### Decision · Program_Chairs · 2026-01-26

Accept (Poster)